# CROSS SPLINE NET: A SIMPLER, MORE INTER-PRETABLE AND UNIFIED MACHINE LEARNING FRAME-WORK

## ABSTRACT

We propose a simple, interpretable and unified machine learning framework called cross spline net (XSN). The framework is built on a combination of spline transformations, projection and cross-network (Wang et al., 2017; 2021). Our framework has a few key contributions. First, XSN is a bounded degree functional ANOVA model, which is simpler, less overfitted, and more interpretable than black box machine learning models. It models feature interactions without parameter or data explosion issue, and it does not require an interaction filtering algorithm like others (Yang et al., 2021; Park et al., 2025) do. Second, XSN tackles the model interpretation problem with pruning and purification algorithms that help users understand the model composition and feature effects. This is an important but challenging problem given the large number of terms and effects after model expansion (Duong et al., 2024). Finally, XSN unifies a variety of machine learning models under the same neural network framework, avoiding some pitfalls (such as being ad hoc, greedy or non-scalable) in the original optimization methods used in these machine learning models. We will use a special type of XSN - TreeNet, to illustrate our point. [1]

## 1 INTRODUCTION

In today's tabular data modeling world, gradient boosting (XGBoost, LightGBM, CatBoost) and fully connected neural network (FCNN) are two of the most popular modeling methods. They are widely used due to their strong performance and ease of use, without the need for careful feature engineering, and the algorithms are effectively parallelized. However, the downside is they are black-box ML models. They are highly complicated, hard to interpret and can be overfitted (with large train-test performance gap) for small datasets. This hinders their application in highly regulated industries like banking. On the other hand, empirical studies suggest that for many tabular data, simpler GAMI-style models (Generalized Additive Model + two-way interaction) can already achieve very good performance (Lou et al., 2013; Dubey et al., 2022), while being more transparent. These GAMI-style models are gaining traction in highly regulated industries; for example, see Hu et al. (2025) and Yang et al. (2021).

In this paper, we propose a new modeling framework that is as performant and easy to use as FCNN or XGBoost, but also simpler, less overfitted and more interpretable. Our framework, cross spline net (XSN), is based on a combination of spline transformations (here we use the term 'spline' loosely to refer to any nonlinear transformation), projection, and cross-network (Wang et al., 2017; 2021). XSN is a simple yet effective design that captures bounded-degree functional ANOVA (FANOVA) models well, including GAMI-style models as a special case. It models a polynomial function on the splines without data or parameter explosion issue or requiring an interaction filtering step (Yang et al., 2021; Park et al., 2025). Second, we develop a comprehensive set of model simplification and interpretability tools to help users understand the XSN model and gain insights into the data. Model interpretation is challenging given the huge number of terms after polynomial expansion, and XSN is filling this important gap. Finally, XSN is a unifying framework. By configuring the

---

[1]The views expressed in this paper are solely those of the authors and do not necessarily reflect the views of their affiliated institutions.

spline layer, we can reproduce or approximate a set of non-neural network models, including linear and polynomial regression models, tree, rule-fit, tree-ensembles (gradient boosting trees, random forest), oblique tree/forests, multi-variate adaptive regression spline (MARS), etc. This unification is both elegant and advantageous. First, neural networks are more extensible than non-neural network models. For example, regularization can be easily added, without changing the training algorithm. Second, it provides free access to scalable, powerful neural network optimization algorithms such as ADAM. This prevents issues like ad-hoc solutions, greediness, or poor scalability found in earlier optimization methods. To summarize, our contribution is threefold:

1. A simple yet effective FANOVA modeling framework that is performant but less overfitted and more interpretable than black-box ML models. It includes the GAMI-styles models as a special case but is not limited to just second order.

2. A comprehensive set of model simplification and interpretability tools which offer insights into model composition and feature effects, from both polynomial regression perspective and functional-ANOVA (FANOVA) perspective.

3. Unification of a set of non-neural network models under the neural network framework, with the benefit of being extensible due to the composability of neural networks and having free access to scalable and powerful optimization algorithms in neural network libraries.

## 2 RELATED WORK

Our work solves two key issues in FANOVA models. First, our method builds higher-order FANOVA models in a scalable way, without the parameter or data explosion issue. Second, we provide a comprehensive set of model simplification and interpretation tools to understand the fitted model, which fills an important gap especially for higher-order models. Finally, it is a unifying framework that sheds light on how conventional modeling problems can be solved in a modern way. We will explain in detail below.

First, the problem of fitting up to second-order FANOVA models is well-studied, but fitting higher-order FANOVA models in a scalable way remains challenging. NAM (Agarwal et al., 2021) and NBM (Radenovic et al., 2022) use a sub-network to fit each main-effect in a generalized additive model (GAM). GAMI-Net (Yang et al., 2021) and NB2M (Radenovic et al., 2022) extend NAM to second order models, where each interacting pair is modeled with a sub-network. Since the number of all variable pairs grows quadratically, they require an interaction screening algorithm like the FAST algorithm in EBM (Lou et al., 2013). Similarly, ANOVA-TPNN (Park et al., 2025) is a tensor product neural network for higher order models, but it also requires an interaction screening algorithm. However, for higher order ($\geq 3$), interaction filtering becomes much more complicated given the combinatorics nature of the problem, and the FAST method (which searches all pairs and all cut-points of each variable exhaustively) in EBM becomes infeasible. To circumvent the parameter explosion problem, CAT (Duong et al., 2024) and SPAM (Dubey et al., 2022) use low-rank approximation to approximate the high dimensional coefficient tensor. Our work provides a new, scalable way to solve the parameter explosion problem using simple yet effective network design. With cross network, each cross layer output ($\boldsymbol{x}^{i+1}$) raises the input ($\boldsymbol{x}^i$) order by one by multiplying with the raw input $\boldsymbol{x}$, i.e., $\boldsymbol{x}^{i+1} = \boldsymbol{x} \odot (W_i \boldsymbol{x}^i + b_i) + \boldsymbol{x}^i$. The weight matrix $W_i$ selects which $i+1$ order interaction to create by crossing $\boldsymbol{x}$ and $\boldsymbol{x}^i$. The number of parameters grows only linearly as the interaction order increases. However, cross network can only model simple interactions like $x_1 x_2$. Combining with spline layer and projection layer, XSN can capture complex $k$-way interactions while being computationally efficient.

The other significant gap we are filling is model interpretation after a FANOVA model is fitted, especially for higher order; without it, the interpretability goal won't be achieved. For GAM and FANOVA models with pre-screened interactions (GAMI-Net, ANOVA-TPNN), interpretation is relatively straightforward. However, without pre-screened interactions, interpretation is much more challenging since one needs to expand the model into additive form and the number of coefficients and functional components can be huge. In Dubey et al. (2022), the authors only explained for an order-2 SPAM-Linear model. But as pointed out in Duong et al. (2024), even for such simple case, its explanation is still too long to be interpretable. Duong et al. (2024) groups variables into a small set of concepts and only explain based on such concepts to avoid the complexity. However, our approach solves this problem head on. Our pruning algorithm effectively reduces the number of

polynomial terms and functional components, and our purification algorithm ensures model components are uniquely identifiable. As for future research, adding regularization can lead to sparsity in the $W_i$ matrices, and making the pruning even more efficient. Note these algorithms are general, they can be used for SPAM or CAT model, and even made into a general-purpose interaction screening algorithm.

Finally, our unifying framework differs from RPN (Zhang, 2024a;b) which is another network unifying SVM, MLP and KAN (Liu et al., 2025). The first difference, as we mentioned, XSN has a comprehensive system of model interpretation tools that RPN does not have. Second, in RPN, the feature transformations are done in a pre-defined way in data expansion step, and only the coefficients on the transformed variables are trained. For example, the Taylor expansion option has a similar goal as cross net to model polynomial functions, but it does so by creating all polynomial bases up to order $d$ which grow exponentially. The authors used a parameter fabrication technique to reduce the number of parameters, but the data expansion step remains inefficient. On the other hand, XSN engineers those polynomial terms as part of the cross network training process which is scalable, and it also trains the spline transformations which can be useful in, e.g., knots selection. Our goal, following the unification under XSN, is to represent as many problems as possible by neural network (with splines and crosses), and solve them in a modern and efficient way.

# 3 CROSS SPLINE NET METHODOLOGY

## 3.1 MODEL FRAMEWORK

Let $y$ be the response and $\boldsymbol{x}$ be the $p$-dimensional feature vector. A variety of machine learning models can be written in the following form:

$$F(\boldsymbol{x}) = \text{Polynomial}(\Phi(\boldsymbol{x})), \tag{1}$$

where $\Phi(\boldsymbol{x})$ is a $l$-dimensional spline transformation function. The spline transformation adds nonlinearity and flexibility, and the polynomial function accounts for feature interactions. Simple examples are linear and polynomial regression ($\Phi(\boldsymbol{x}) = \boldsymbol{x}$). Regression tree and rule-fit also fall into this category, where $F(\boldsymbol{x}) = \sum_i a_i \left( \prod_{j \in \text{Path}(i)} \mathbb{I}(x_j \lessgtr c_{ij}) \right)$, $\Phi(\boldsymbol{x}) = \{\mathbb{I}(x_j \lessgtr c_{ij})\}$, $a_i$ is the node value for leaf node $i$ in the tree, $x_j$ is one of the splitting variables leading to leaf node $i$, $\mathbb{I}()$ is the indicator function and $c_{ij}$ is the split point. Tree ensemble (XGBoost, random forest) has a similar form but is much more complicated. For oblique tree/forests, each split is on a linear projection of the predictors (Breiman, 2001; Rodríguez et al., 2006), hence $\Phi(\boldsymbol{x}) = \{\mathbb{I}(\boldsymbol{w}_{ij}^T \boldsymbol{x} \lessgtr c_{ij})\}$. For MARS (Friedman, 1991), $\Phi(\boldsymbol{x}) = \{(x_j - c_{ij})_+, (c_{ij} - x_j)_+\}$.

Cross spline net fits the class of models in Equation (1) in a unified and natural way. Firstly, it is easy to implement spline functions in a neural network. The hinge function used in MARS is just the RELU activation. The indicator function $\mathbb{I}(x \lessgtr c)$ can be well approximated by a sigmoid activation function $\sigma(\alpha + \beta x)$, with a large, fixed value of $\beta$ and an appropriate value of $\alpha$ (see Fig. 1(a)). Similarly, the indicator function $\mathbb{I}(\boldsymbol{w}^T \boldsymbol{x} \lessgtr c)$ can be well approximated by $\sigma(\alpha + \boldsymbol{w}^T \boldsymbol{x})$. The total number of splines can be large, so we add a projection layer to select important splines. This also makes XSN more computationally efficient and less overfitting. Finally, the polynomial function is modeled via cross layers proposed in Wang et al. (2017; 2021), where stacking $k$ cross layers results in a $k + 1$ degree polynomial. Importantly, the number of parameters only grows linearly when the interaction order increases. A special XSN with sigmoid spline, TreeNet, is displayed in Fig. 1(b).

Note our proposed cross spline network has a key difference from the Deep & Cross network (DCN, DCN-v2) in Wang et al. (2017; 2021). Since the cross layer can only model polynomials, the authors supplemented it with a deep FCNN after (or alongside) the cross layers to capture nonlinearity. However, our way of handling nonlinearity with spline transformation before the cross layers is the key to XSN's better model interpretability and the unification of models in Equation (1). With FCNN being entangled with cross-net, DCN and DCN-v2 are still black-box. On the other hand, XSN is less complicated and more interpretable when the polynomial function has a low degree.

The unification of model classes in Equation (1) under XSN brings some new advantages. The first one comes from the optimization method. Specifically, the current optimization methods for these models are case specific, can be greedy, ad-hoc or non-scalable to large data, as opposed to the

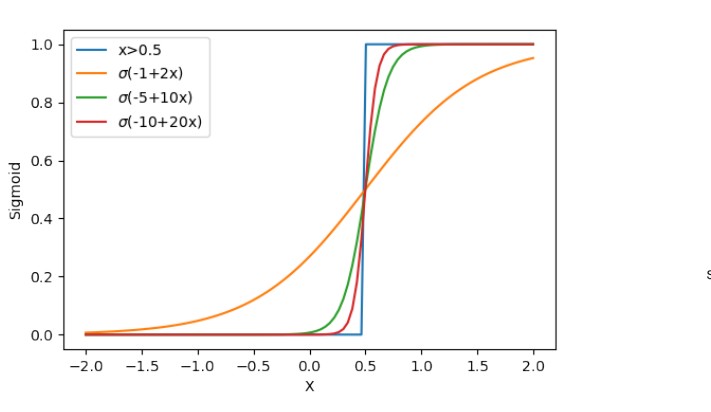 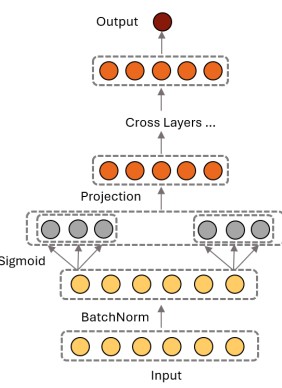

(a) Sigmoid function with varying scale and center.

(b) Architecture of TreeNet

Figure 1: Sigmoid basis function plots (left) and architecture for TreeNet (right).

general, scalable and powerful neural network optimization methods. For example, the optimization methods for tree, oblique tree, XGBoost, Random Forest, MARS, and so on are all complex and need to be developed individually by expert programmers. Moreover, their optimization algorithms can be greedy or even ad-hoc. While they work well in general, it can sometimes be hard to find a good solution. For example, tree is a greedy algorithm as it only considers one split at a time. This means the splits are not optimized jointly. In particular, the root node is determined by main effects only, without taking into consideration the interaction effects. This makes it very hard to capture pure interaction effects when the model does not have any main effects, i.e., $\mathbb{E}(F(\boldsymbol{x})|x_j) = 0$ for all $j$. MARS employs a greedy forward selection algorithm and is slow since it scans each unique value of each predictor for potential knots. Oblique-tree/forest often tries a few random projections since they cannot optimize the projection direction $\boldsymbol{w}$ easily. On the other hand, with advanced neural network optimization routines such as ADAM, XSN has a better chance of finding the best solution. This is especially true for oblique tree/forests, since projections are easily optimized in neural networks, but it is also true for tree/tree-ensemble where "splits" can be optimized jointly in TreeNet (see pure interaction case in Sec. 4). Since neural network optimization routines are all mini-batched and parallelized, they are also scalable to large data. A comparison of run time among TreeNet, FCNN and XGBoost is provided in Appendix A. We can see TreeNet is comparable with FCNN and XGBoost for 100K case and much faster than XGBoost for 1M case (since XGBoost takes many more iterations to converge). It is also much faster than alternative interpretable models like GAMI-Net and TPNN due to its efficient design.

Another benefit is XSN can mimic but avoid some drawbacks of the non-neural network models. For example, trees use the discontinuous indicator function as basis. It requires multiple splits to model any continuous functions, which leads to jumpiness and overfitting. However, sigmoid basis can approximate both linear and indicator functions, making it more versatile (see Fig. 1(a)). XSN with sigmoid basis mimics tree but is smoother and less overfitted. For this reason, we call it TreeNet and it will be our focus in the rest of this paper.

### 3.2 A SPECIAL CASE: TREENET

TreeNet is aimed at mimicking tree/tree-ensembles due to their popularity. It is XSN with sigmoid basis $(\sigma(\alpha + \beta x))$. The architecture of TreeNet is shown in Fig. 1(b). Specifically, we first apply batch normalization to the input, to avoid the vanishing gradient issue related to large $x$ values. Then we create $m$ sigmoid basis for each input variable $x_j$, i.e. $\{\sigma(\alpha_{ij} + \beta_{ij}x_j), i = 1, \ldots, m; j = 1, \ldots, p\}$. There are a total of $mp$ bases. Not all bases will be important, so we insert a linear projection layer to reduce the dimension down to $d$. Next, we stack $k$ cross layers (Wang et al. (2017; 2021)) to model the polynomial function and finally output. The value $k$ determines the order of interactions in TreeNet. The maximum order of interactions modeled is $k + 1$. The hyper-

parameter $k$ plays a similar role as the depth of tree in tree-ensembles. Note that in FCNN, it is impossible to limit the order of interactions.

The hyper parameters $m, d$ and $k$ can be tuned, and we provide a searching grid in Appendix B. However, there are also good default values. For $m$, since each sigmoid basis can approximate a local linear segment and the "knots" are optimized by the weights ($\alpha$'s and $\beta$'s), we only need a small number of bases. Default $m = 5$ is sufficient since practical data don't have many cycles. For $d$, it grows with the intrinsic dimension of the data. We experimented with $d = 20$ in both our simulation studies and real data cases, and it worked well. For $k$, the order of interaction is typically low for tabular data. Studies suggest that a second order model ($k = 1$) can already fit very well to the data. Here we recommend $k = 2$ to fit up to 3-way interactions.

For the optimizer, by default, we use ADAM (Kingma & Ba, 2014), with a learning rate of 0.02, batch size 1% of sample size, and learning rate decay of 0.995 per epoch. The best number of epochs is determined by early stopping. We call this TreeNet with the above default hyper-parameter settings TreeNet2 (see Appendix B for details). As we will see in Sec. 4 and Sec. 5, TreeNet2 works well in general and can be used as an off-the-shelf machine learning algorithm. Only exception we see is when the data has many features but not enough observations. In this case, TreeNet supports regularization and a regularizer can be added to the projection layer to control overfitting (see Sec. 5). Finally, we provided a parameter ablation study for three key hyper-parameters in Appendix C, to show the robustness of TreeNet2 setting.

### 3.3 MODEL INTERPRETABILITY

Another key contribution of this paper is interpretability. We interpret the XSN model from two angles. First, we can extract all coefficients of the modeled polynomial function in XSN, and interpret the important terms. Second, we can group different terms and interpret the model from a functional-ANOVA (FANOVA) perspective, a common approach for GAMI-style models (Hu et al., 2025).

Given a fitted XSN model with $k$ cross layers and $l$ bases $(\Phi_1(\boldsymbol{x}), \ldots, \Phi_l(\boldsymbol{x}))$ after the spline transformation (but before projection layer), the total number of polynomial terms is $O(l^{(k+1)})$ which can be huge. Most of the terms will be unimportant. To quantify the importance, we use $L_1$-norm. Specifically, for a given polynomial term, $\gamma_{\mathcal{S}} \prod_{j \in \mathcal{S}} \Phi_j(\boldsymbol{x})$, where $\mathcal{S} \subseteq \{1, 2, \ldots, l\}$ and $\gamma_{\mathcal{S}}$ is the coefficient (see Appendix D for how to extract the coefficients), its $L_1$-norm is $\mathbb{E}_{P_{\mathbf{X}}} |\gamma_{\mathcal{S}} \prod_{j \in \mathcal{S}} \Phi_j(\boldsymbol{x})|$, $\boldsymbol{x} \sim P_{\mathbf{X}}$. Evaluating the $L_1$-norm for all the terms can be time consuming. Therefore, we propose a numerically efficient method based on Holder Inequality: $\mathbb{E}_{P_{\mathbf{X}}} |\prod_{j \in \mathcal{S}} \Phi_j(\boldsymbol{x})| \leq \prod_{j \in \mathcal{S}} (\mathbb{E}_{P_{\mathbf{X}}} |\Phi_j(\boldsymbol{x})|^{|\mathcal{S}|})^{1/|\mathcal{S}|}$. This provides an upper bound for the $L_1$-norm. Once each $\mathbb{E}_{P_{\mathbf{X}}} |\Phi_j(\boldsymbol{x})|^{|\mathcal{S}|}$ is calculated[2] (which is fast since there are only $l$ bases), the upper bound can be computed quickly as a scalar product. We can rule out most of the trivial terms based on upper bounds and only compute the $L_1$-norm for the remaining terms. This greatly reduces the computation time. For the top important polynomial terms, we can inspect their composition, i.e., which spline bases are involved and how each spline basis looks like.

Our second interpretation approach is from the FANOVA perspective. The FANOVA framework decomposes a complex high-dimensional function into lower order effects:

$$F(\boldsymbol{x}) = \gamma_0 + \sum_j f_j(x_j) + \sum_{j<k} f_{jk}(x_j, x_k) + \cdots \tag{2}$$

Lower order FANOVA models such as GAMI-style models have recently become popular (Hu et al., 2025) due to their interpretability and good model performance. In the XSN framework, if the spline transformations are chosen to be univariate (e.g. TreeNet, where $\{\Phi_1(\boldsymbol{x}), \ldots, \Phi_l(\boldsymbol{x})\} = \{\sigma(\alpha_{ij} + \beta_{ij}x_j), i = 1, \ldots, m; j = 1, \ldots, p\}$), it automatically falls into the FANOVA framework. In particular, if the number of cross-layers is one, it is a main-effect plus two-way interaction model, just like the GAMI-class models.

To obtain the FANOVA representation, one only needs to sum up the polynomial terms (that remains after filtering) which relate to the same variable/set of variables. For example, assume

---

[2]Calculation can be done on a sample using Monte-Carlo method.

$\gamma_1 \sigma(\alpha_{11} + \beta_{11} x_1)$, $\quad \gamma_2 \sigma(\alpha_{12} + \beta_{12} x_1)$, $\quad \gamma_3 \sigma(\alpha_{11} + \beta_{11} x_1) \times \sigma(\alpha_{12} + \beta_{12} x_1)$, $\quad \gamma_4 \sigma(\alpha_{21} + \beta_{21} x_2)$, $\quad \gamma_5 \sigma(\alpha_{11} + \beta_{11} x_1) \times \sigma(\alpha_{12} + \beta_{12} x_2)$ are the remaining polynomial terms after filtering. The first three terms all capture part of the main-effect of $x_1$, the fourth term captures the main-effect of $x_2$ and the last term captures the interaction-effect of $x_1, x_2$. The component $f_1(x_1)$ is simply the sum of the first three terms. However, XSN does not do interaction filtering like the GAMI-style models, thus there can be many small, false interaction effects. Therefore, we implement a model pruning algorithm in Algorithm 1 to keep only the top $K$ components depending on performance degradation tolerance. Note the filtering threshold $\epsilon$ should be chosen small (e.g., 0.001) as important effect can be split among different terms, causing each of them to be small.

---

**Algorithm 1** Pruning algorithm for XSN

---

**Require:** Dataset $\mathcal{D} = \{(\boldsymbol{x}_1, \boldsymbol{x}_2, \ldots \boldsymbol{x}_n), (y_1, y_2, \ldots y_n)\}$, where $\boldsymbol{x}_i$ is $p$-dimensional feature; fitted XSN model with univariate spline transformation; small threshold $\epsilon$ for filtering polynomial terms and tolerance $\tau$ for performance degradation.

1: Filter the polynomial terms based on $L_1$-norm, i.e., only keep terms with $L_1$-norm $\geq \epsilon$.

2: For the retained terms, sum up the terms related to the same variable/set of variables. This gives the functional components $f_\mathcal{S}(\boldsymbol{x}_\mathcal{S})$, $\mathcal{S} \subseteq \{1, 2, \ldots, p\}$. Let the number of components be $K_{\max}$.

3: Rank the functional components by importance (using $L_1$-norm, or the drop in model performance when removing this component).

4: **for** $K = 1$ to $K_{\max}$ **do**

5:     Refit the response using the top $K$ most important components with linear (continuous) or logistic (binary) regression.

6:     Stop when the model performance degradation is smaller than threshold $\tau$.

7: **end for**

8: **return** Top $K$ components $\{f_\mathcal{S}(\boldsymbol{x}_\mathcal{S}), \mathcal{S} \in \mathcal{G}\}$ with associated polynomial terms.

---

After pruning, we get the top $K$ functional components. When there are both main-effects and interactions, the functional representation is not unique, unless hierarchical orthogonality is imposed (Hooker, 2007). To enforce hierarchical orthogonality, purification is usually done to remove the lower order effects from the higher order term. One such algorithm in proposed in Lengerich et al. (2020). Here we propose a new purification algorithm using TreeNet to remove lower order effects, in Algorithm 2. It is a waterfall algorithm which purifies from higher order effects to lower orders. After purification, the importance should be recalculated based on the purified effects and the lower-order effects (1-d and 2-d) can be interpreted.

---

**Algorithm 2** Purification algorithm for XSN

---

**Require:** Dataset $\mathcal{D} = \{(\boldsymbol{x}_1, \boldsymbol{x}_2, \ldots \boldsymbol{x}_n), (y_1, y_2, \ldots y_n)\}$, $K$ top functional components $\{f_\mathcal{S}(\boldsymbol{x}_\mathcal{S}), \mathcal{S} \in \mathcal{G}\}$ with highest order of interaction $d_{\max}$.

1: **for** $d$ in $d_{\max}$ to 2 **do**

2:     **for** $\mathcal{S} \in \mathcal{G}$ where $|\mathcal{S}| = d$ **do**

3:         **Fit:** fit a TreeNet $T(\boldsymbol{x}_\mathcal{S})$, with $d - 2$ cross layers, to $f_\mathcal{S}(\boldsymbol{x}_\mathcal{S})$ with the set of variables $\boldsymbol{x}_\mathcal{S}$.

4:         **Purify:** $f_\mathcal{S}^{\text{pure}}(\boldsymbol{x}_\mathcal{S}) = f_\mathcal{S}(\boldsymbol{x}_\mathcal{S}) - T(\boldsymbol{x}_\mathcal{S})$. This removes all lower order effects from $f_\mathcal{S}(\boldsymbol{x}_\mathcal{S})$.

5:         **Decompose:** $T(\boldsymbol{x}_\mathcal{S}) = \sum_{\mathcal{R}:\{\mathcal{R} \subset \mathcal{S}, |\mathcal{R}| < d\}} g_\mathcal{R}(\boldsymbol{x}_\mathcal{R})$ by combining polynomial terms. Here no pruning is needed since $d$ is small.

6:         **for** $\mathcal{R} \subset \mathcal{S}, |\mathcal{R}| < d$ **do**

7:             **Update:** $f_\mathcal{R}(\boldsymbol{x}_\mathcal{R}) \leftarrow f_\mathcal{R}(\boldsymbol{x}_\mathcal{R}) + g_\mathcal{R}(\boldsymbol{x}_\mathcal{R})$. This keeps the total effects unchanged.

8:             **Update:** $\mathcal{G} \leftarrow \mathcal{G} \cup \{\mathcal{R}\}$ if $\mathcal{R} \notin \mathcal{G}$

9:         **end for**

10:     **end for**

11: **end for**

12: **return** Purfied functional components $f_\mathcal{S}^{\text{pure}}(\boldsymbol{x}_\mathcal{S})$.

---

Our pruning and purification algorithms are efficient. In Appendix E, we show the interpretation run time for a fitted TreeNet2 model with more than half million terms. The interpretation time varies from less than 0.5 minutes to 1.5 minutes, depending on the sample size and number of components.

## 4 SIMULATION EXPERIMENT

In this section, we run simulation studies to compare TreeNet with XGBoost and FCNN. We look at main-effect only, two-way interaction and three-way interaction cases. For each scenario, we have different flavors: continuous form, jumpy form (with indicator function), and pure interaction form ($\mathbb{E}(F(\boldsymbol{x}) \mid x_j) = 0$ for all $j$). We simulate 30 predictors from uniform$(-1, 1)$ distribution and response $y = F(\boldsymbol{x}) + \epsilon, \epsilon \sim N(0, 1)$. A total of 10,000 observations are simulated for each functional form, divided into 70% training and 30% validation data. A separate test set with 50,000 observations is used to assess the test performance. The details are given in Appendix F.

We fit five algorithms to our data: TreeNet, XGBoost, FCNN, TreeNet2 and XGBoost3. The first three algorithms are tuned to achieve optimal performance. The last two algorithms come with fixed hyper-parameters except the number of epochs/trees which is determined with early stopping. For TreeNet2, it models up to three-way interactions. As a comparison, we fit XGBoost3, where max tree depth is fixed at 3. The details about the hyper-parameter settings can be found in Appendix B.

We ran our experiment 10 times with different random data generation seeds. The test performances for all five algorithms and eight cases are shown in Table 8, Appendix F, where the average MSEs and the standard deviations (in parenthesis) are reported. The average test MSEs and the train-test performance gaps are plotted in Fig. 2 (where 1_cont means main_cont, 2_cont means 2way_cont).

First, focusing on the testing performance, we see that TreeNet performs best in all cases except for the jumpy cases: main_jump, 2way_jump, and 3way_jump, where XGBoost is the best. This is expected because the underlying functions in these cases contain indicator functions, where XGBoost can perform very well with its binary split. However, there is not much difference in the test MSEs between TreeNet and XGBoost for the main_jump ($\sim 0.02$) and 2way_jump ($\sim 0.07$) cases. For the 3way_jump case, as we keep adding more jumpy interactions, the difference increases to 0.2. This indicates that TreeNet provides a reasonable approximation to jumpy functions, and the loss in performance is small when the number of jumpy functions is small.

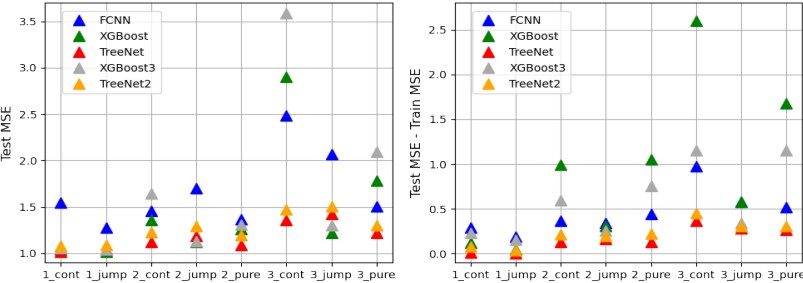

Figure 2: Test MSE (left) and train-test performance gap (right) for five algorithms in eight cases. TreeNet and TreeNet2 show good performance and small train-test gap

XGBoost performs worse than TreeNet except for the three jump cases. It does especially poorly for 3way_cont (test MSE 2.9) and 3way_pure (test MSE 1.79). One reason is that XGBoost is less suitable for modeling smooth functions. The other reason, as explained in Sec. 3 is that pure interactions do not have any lower-order effects. They are hard to capture for the greedy tree splitting algorithm which is driven by lower-order effects at the top layers of the tree. This drawback is exacerbated with shallow trees, as we see from XGBoost3 model in the 3way_pure case. However, TreeNet optimizes all parameters jointly, so it does not have this issue.

For FCNN, it performs the worst in all cases except 3way_cont and 3way_pure, where it does better than XGBoost but worse than TreeNet. It is not good at capturing jumpy patterns and does worse than TreeNet and XGBoost; also, it does not do well in capturing some low-dimensional smooth functions. For example, it does not do well for main_cont, with test MSE significantly higher than TreeNet and XGBoost.

For TreeNet2 and XGBoost3, we can see TreeNet2 performs well, and is only slightly worse than the tuned TreeNet. It does worse than XGBoost3 for 2way_jump and 3way_jump, comparable or better for the other cases. It is much better especially for 3way_cont and 3way_pure.

Turning to the train-test performance gap (Fig. 2), we see that TreeNet and TreeNet2 have the smallest gap. This confirms our theory that TreeNet overfits less than XGBoost, since its basis functions are continuous. As for FCNN, it inherently models high-dimensional interactions, making it more prone to overfitting than TreeNet and less suitable for low-dimensional models.

When the sample size increases, all algorithms perform better, with FCNN which showed the worst performance previously, seeing the most improvement. As an illustration, we run the simulation study for a sample size of 50K (train plus validation), for the 3way cases. The test MSEs (averaged over 10 trials) are shown in Table 1, where the numbers in parenthesis are the test MSEs from the 10K case. We can see TreeNet has a close performance to XGBoost for the 3way_jump case (better approximation to indicator function), and still does significantly better than XGBoost in the other two cases. TreeNet2 does well in all cases and only sees a tiny performance loss compared to TreeNet. XGBoost3 only does well for the jump cases and shows a significant performance loss compared to TreeNet or TreeNet2 for the other two cases.

Table 1: Test MSE for 50K samples averaged over 10 trials, compared to 10K case (in parenthesis).

| Scenario | Train MSE | | | | | Test MSE | | | | |
|---|---|---|---|---|---|---|---|---|---|---|
| | TreeNet | FCNN | XGBoost | TreeNet2 | XGBoost3 | TreeNet | FCNN | XGBoost | TreeNet2 | XGBoost3 |
| 3way_cont | 1.02 | 1.03 | 0.49 | 1.00 | 2.24 | 1.09 | 1.19 | 1.89 | 1.10 | 2.84 |
| | (1.00) | (1.51) | (0.30) | (1.02) | (2.44) | (1.36) | (2.48) | (2.90) | (1.47) | (3.58) |
| 3way_jump | 1.10 | 1.06 | 0.83 | 1.06 | 1.01 | 1.18 | 1.15 | 1.08 | 1.19 | 1.16 |
| | (1.14) | (1.49) | (0.64) | (1.20) | (0.96) | (1.42) | (2.07) | (1.22) | (1.50) | (1.30) |
| 3way_pure | 1.00 | 1.00 | 0.34 | 1.00 | 1.36 | 1.05 | 1.10 | 1.30 | 1.06 | 1.79 |
| | (0.96) | (0.98) | (0.11) | (0.99) | (0.94) | (1.22) | (1.50) | (1.79) | (1.30) | (2.09) |

## 5 REAL DATA EXPERIMENT

In this section, we carry out experiments on seven public datasets: Bike sharing (Fanaee-T, 2013), California housing (Pace & Barry, 1997), MSLR (Qin & Liu, 2013), HELOC (FICO, 2018), Spambase (Hopkins et al., 1999), Magic (Bock, 2004) and Madelon (Guyon, 2004). The first three have continue response and the latter four have binary response. Details on the datasets is provided in Appendix G.

Table 2: Performance for public data. Performance averages and standard deviations (in parenthesis) over 10 trials are reported

| Data | Metric | TreeNet | FCNN | XGBoost | TreeNet2 | XGBoost3 | GAMINet | TPNN2 |
|---|---|---|---|---|---|---|---|---|
| BikeShare | Train MSE | 0.090 (0.012) | 0.074 (0.009) | 0.042 (0.008) | 0.098 (0.006) | 0.079 (0.003) | 0.153 (0.043) | 0.183 (0.032) |
| | Test MSE | 0.114 (0.005) | 0.119 (0.003) | 0.104 (0.003) | 0.115 (0.004) | 0.110 (0.004) | 0.157 (0.041) | 0.190 (0.025) |
| CalHousing | Train MSE | 0.246 (0.021) | 0.206 (0.017) | 0.037 (0.014) | 0.252 (0.021) | 0.119 (0.002) | 0.313 (0.025) | 0.265 (0.015) |
| | Test MSE | 0.275 (0.012) | 0.272 (0.014) | 0.213 (0.007) | 0.276 (0.011) | 0.226 (0.007) | 0.323 (0.024) | 0.281 (0.011) |
| MSLR | Train MSE | 0.533 (0.011) | 0.531 (0.004) | 0.469 (0.025) | 0.538 (0.010) | 0.542 (0.001) | 0.571 (0.004) | 0.558 (0.004) |
| | Test MSE | 0.543 (0.004) | 0.551 (0.001) | 0.530 (0.004) | 0.548 (0.006) | 0.548 (0.001) | 0.572 (0.005) | 0.561 (0.003) |
| HELOC | Train AUC | 0.809 (0.004) | 0.811 (0.006) | 0.900 (0.023) | 0.806 (0.003) | 0.834 (0.006) | 0.804 (0.003) | 0.800 (0.007) |
| | Test AUC | 0.797 (0.010) | 0.790 (0.008) | 0.789 (0.009) | 0.796 (0.010) | 0.796 (0.008) | 0.799 (0.008) | 0.793 (0.009) |
| Spambase | Train AUC | 0.990 (0.003) | 0.993 (0.002) | 0.999 (0.001) | 0.987 (0.003) | 0.997 (0.001) | 0.986 (0.002) | 0.980 (0.010) |
| | Test AUC | 0.981 (0.003) | 0.978 (0.003) | 0.986 (0.003) | 0.980 (0.003) | 0.985 (0.002) | 0.977 (0.003) | 0.974 (0.012) |
| Magic | Train AUC | 0.947 (0.003) | 0.946 (0.003) | 0.989 (0.005) | 0.947 (0.005) | 0.975 (0.004) | 0.925 (0.005) | 0.907 (0.010) |
| | Test AUC | 0.935 (0.005) | 0.928 (0.005) | 0.933 (0.005) | 0.935 (0.004) | 0.933 (0.005) | 0.922 (0.007) | 0.907 (0.010) |
| Madelon | Train AUC | 0.888 (0.021) | 0.866 (0.032) | 1.000 (0.000) | 0.855 (0.090) | 0.992 (0.010) | 0.879 (0.024) | 0.657 (0.02) |
| | Test AUC | 0.828 (0.022) | 0.628 (0.032) | 0.881 (0.047) | 0.819 (0.088) | 0.808 (0.022) | 0.840 (0.030) | 0.615 (0.01) |

We divide each dataset into training (50%), validation (25%) and testing (25%) samples. In addition to FCNN and XGBoost, we also compare TreeNet with two alternative interpretable models, GAMI-Net and second-order TPNN (TPNN2). We use the training and validation sets to tune the models and test set to assess the performance. Details on the model tuning is provided in Appendix B. Again, we run the experiment 10 times with different data splitting seeds. The average test performances and standard deviations (in parenthesis) are shown in Table 2. We can see TreeNet and TreeNet2 again show robust model performance with small overfitting gap. The performance is

comparable with XGBoost on all datasets except for California housing and Madelon, where XG-Boost is significantly better than all other models. For California housing, we suspect this is similar to the jumpy case in our simulation study, where the underlying data pattern is jumpy and more suitable for XGBoost model. For Madelon, this is a 'wide' data as it has many features (500) but small number of observations and can overfit easily. TreeNet didn't do well initially (with test AUC around 0.63) since after spline transformation, the number of bases can exceed sample size and over-fits. Therefore, we added an L0 regularizer to the projection layer, which penalizes the (thousands of) spline bases. This effectively mitigates the overfitting issue, and the model performance is much better. Note for all other datasets, regularization is not needed even for MSLR which has 136 features but sufficient sample size. Finally, comparing TreeNet with the other two interpretable models (GAMI-Net and TPNN2), TreeNet has better performance for most cases, except for Madelon where GAMI-Net is better.

## 6 INTEPRETABILITY ANALYSIS

We illustrate the interpretability methods proposed in Sec. 3.3 using one simulated data (3way_cont) and one real data (bike sharing). For both datasets, we fit off-the-shelf TreeNet2 with default hyper-parameter settings. The true model form for 3way_cont is shown below for reference:

$$f(x) = x_1 + 2x_2^2 + 2(1 + x_3)^{1/3} + 2x_4(x_4 > 0) + \sin(2\pi x_5) + e^{x_6} + 2x_1x_2 + 2\sin(\pi(x_3 + x_4))$$
$$+ 2|x_5x_6| + 3x_1e^{|x_2x_3|} + 3x_5\sin(\pi(x_4 + 1.5x_6))$$

For the 3way_cont data, there are 30 variables, hence 150 sigmoid bases in TreeNet2. Counting all the polynomial terms (up to third order), the total number of coefficients is 585,276. This is a lot, so we apply our model pruning algorithm. After the $L_1$-norm filtering with a small threshold of 0.001, the number of coefficients reduces significantly to around 12,000. Grouping by functional components, these 12K coefficients belong to 982 functional components. Most of these functional components are insignificant. Figure 3(a) shows the top 14 functional components and the validation MSE when refitting with the top components. We can see the MSE curve becomes flat after the $x_1 \times x_3$ interaction. Therefore, one can prune away all the functional components after $x_1 \times x_3$. After pruning, the remaining number of coefficients reduces to 701, and the test MSE improves to 1.27 (compared to 1.46 for the full TreeNet2).

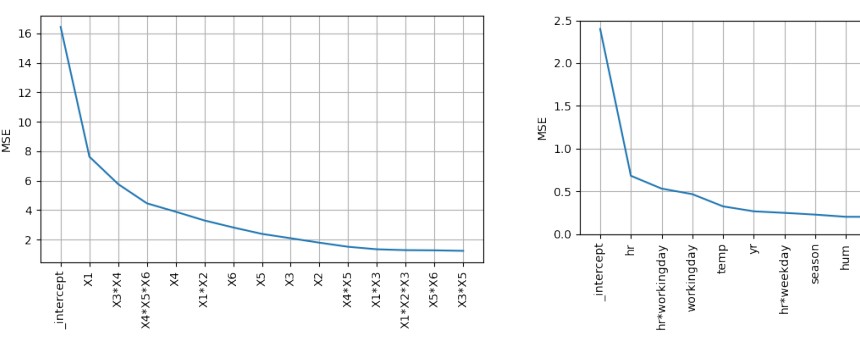

(a) MSE curve for model pruning (3way_cont).     (b) MSE curve for model pruning (Bike sharing).

Figure 3: Model pruning plot. Validation MSE vs number of top components retained.

The list of top functional components is as expected. None of the noise variables appear in the top list. Some interactions like $x_1 \times x_3$ occur because the higher order term $3x_1e^{|x_2x_3|}$ is not pure, and it can project onto lower order effects. On the other hand, this makes the 3-way interaction among $x_1, x_2$ and $x_3$ weaker. This ties to the fact that FANOVA decomposition is not unique without orthogonality constraint, as lower order effects can be moved in and out of a higher order effect. Our purification algorithm imposes orthogonality and ensures no confounding between low and high order effects. In this case, the correlations among the unpurified functional components are small (the largest correlation is -0.17 between $x_3$ and $x_1 \times x_3$), hence the purification does not have too much impact, and we don't show the details here.

For the bike sharing data, there are 11 variables. Three of them are binary, so no spline transformation is needed. In total, there are 15,180 coefficients. We apply the steps in the model pruning algorithm. After $L_1$-norm filtering with a small threshold of 0.001, the number of coefficients reduces substantially to 4,059. Grouping by functional components, these 4K coefficients belong to 226 functional components. Using 2% as tolerance for performance degradation, the pruning algorithm stops after 13 top components have been added. The test MSE for the pruned model is 0.136, a little worse than the full TreeNet2 model (0.115), but the number of retained coefficients is only 582. The pruned model is much simpler. Fig. 3(b) shows the top 13 functional components and the validation MSE when refitting with the top components. The list of top functional components includes well-known main effects like *hr*, *temp*, *workingday*, and well-known interactions like *hr×workingday*. However, these functional components have not been purified, so the higher and lower order components can be confounded. The correlation among the top 13 components is shown in Fig. 6, Appendix H. We can see there exists some high correlations between interaction and main effects, for example, *hr×workingday* and *workingday* have a negative correlation -0.4; *season×temp* and *temp* have a positive correlation of 0.6. This can lead to identification of false interactions. By applying our purification algorithm, the correlation among higher order and lower order effects reduces to $\approx 0$ (Fig. 7). This provides a more accurate view of the true interaction effects in the data. The new ranking of the purified components is provided in Fig. 8. The effect for *workingday*, after purification, becomes quite small.

Finally, the purified 1-d and 2-d effects can be visualized. Fig. 4 shows the most important two-way interaction (*hr×workingday*) and the corresponding two main effects. From *hr*, we can see bike rental counts have two peaks during the morning and afternoon rush hours and one valley around 4 am. This makes sense since some people use bikes to commute. For *workingday*, it is almost flat (consistent with low importance in Fig. 8), indicating no overall bike rental count difference among working day and non-working day. However, the interaction plot tells an interesting story about its effect. On workingday, we observe the two peaks around morning and afternoon rush hours. However, on non-workingday, we see peaks at two different times, one around 3am and one around 11am. This suggests there are more bike rentals after midnight and in the late morning on non-working days, as compared to working days. This reveals a distinct pattern in bike rental behavior when used for commuting or leisure.

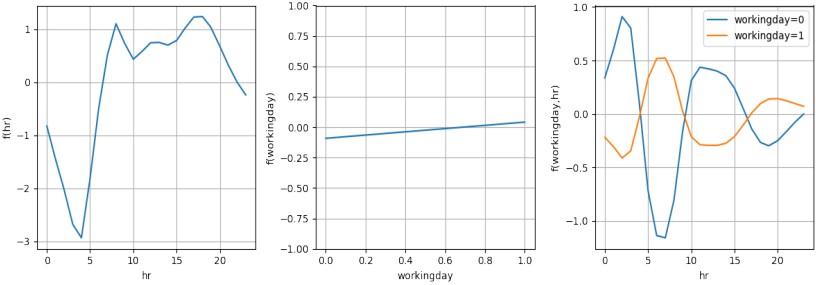

Figure 4: Plot for selected 1-d and 2-d components, after purification.

## 7 CONCLUSION

In this paper, we proposed a simple yet effective network design, XSN, to model bounded-degree FANOVA models in a scalable way and interpret the model with comprehensive pruning and purification algorithms. While we focused on TreeNet, the framework is much broader and can be further extended to other cases. The key ingredient of XSN is spline transformation (for which we have numerous choices) and feature multiplication. While we use cross-net for feature multiplication, we can replace it with a more structured cross network to make it even simpler and more transparent. Regularization can also be added to create a more parsimonious model. This makes it easy to extend the XSN framework to other cases, an area we are actively exploring.

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

# A RUN TIME COMPARISON

Here we compare the run time among TreeNet, FCNN, XGBoost, GAMI-Net and TPNN2. FCNN and XGBoost are known to be well parallelized and highly efficient. For the comparison, we simulate data from 3way_cont example, with different sizes (100K and 1 Million) and number of variables (30 and 100). Each data is divided into 70% training and 30% validation and the models are trained on the training set and evaluated on the validation set. We compare TreeNet, FCNN, XGBoost, GAMI-Net and TPNN2 with the following selected hyper-parameter setting (which we think are reasonable):

- **TreeNet**: we use default setting as in TreeNet2 (with 2 cross layers, 20 neurons in the projection layer and 0.02 learning rate).

- **FCNN**: we use a fixed setting with hidden layer structure [100, 50, 25] , learning rate 0.001 and batch size 1% of training sample.

- **XGBoost**: we use a fixed setting with max depth 5, learning rate 0.05, subsample 0.7 and reg_lambda 1.

- **GAMI-Net**: we use a fixed setting with learning rate 0.005, interaction number 20, and subnets structure [20,20,20].

- **TPNN2**: we use a fixed setting with learning rate 0.01, batch size 1% of training sample, number of K=10, and number of interactions 20.

We run our experiment on a machine with 500G RAM and a 72-core Intel(R) Xeon(R) CPU at 2.60GHz speed. TreeNet, FCNN, GAMI-Net and TPNN2 are trained for up to 400 epochs until validation MSE stops improving, and XGBoost is trained for up to 2000 trees until convergence. Table 3 shows the run time for model fitting, per iteration (epoch or tree) and total time.

First, comparing TreeNet with FCNN and XGBoost, we can see XGBoost is the fastest in terms of time per iteration; however, it generally takes much more iterations to converge than FCNN or XSN. Of course, the number of iterations needed depends on the learning rate. In our setting, the learning rate of XGBoost (0.05) is larger than that of FCNN or XSN, and is in line with the recommended value from the literature (between 0.01 and 0.1). For total run time, we can see the speed of XSN is comparable with FCNN and XGBoost for 100K case, and XGBoost is much slower in the 1M case since it requires many more iterations to converge. The reason that FCNN and XSN require less iterations to converge can be due to mini-batching. With 1% batch size, FCNN and XSN makes 100 parameter updates per epoch, while XGBoost only makes one parameter update per tree. This can be seen from 5, where the validation loss decreases much faster for FCNN and XSN. Comparing FCNN with XSN, XSN takes more time per epoch, especially when $p$ is 100. This is expected because the spline transformation creates 500 basis, compared to only 100 hidden neurons in FCNN. However, the time per epoch is only 2X of FCNN because (1) the connection between spline layer and input layer is very sparse. One input variable is only connected to 5 basis, and the number of weight parameters for the spline layer is only 500 (2) the projection layer further reduces the dimension to only 20. In terms of total run time, FCNN and XSN are comparable.

Second, comparing TreeNet with alternative interpretable models GAMI-Net and TPNN2, TreeNet is much faster than both. In particular, the speed of TPNN2 is quite slow, and can be order of magnitudes slower than TreeNet. This shows the benefit of the simple and efficient design XSN has.

Table 3: Run time per iteration (tree/epoch) and total run time.

| | | TreeNet | | | FCNN | | | XGBoost | | | GAMI-Net | | | TPNN2 | | |
|---|---|---|---|---|---|---|---|---|---|---|---|---|---|---|---|---|
| $n$ | $p$ | Train (s/iter) | iter | Train (total) | Train (s/iter) | iter | Train (total) | Train (s/iter) | iter | Train (total) | Train (s/iter) | iter | Train (total) | Train (s/iter) | iter | Train (total) |
| 100K | 30 | 0.38 | 104 | 40 | 0.25 | 204 | 51 | 0.018 | 1785 | 32 | 0.7 | 212 | 148 | 13.3 | 140 | 1857 |
| 100K | 100 | 0.60 | 159 | 96 | 0.26 | 192 | 51 | 0.049 | 1170 | 57 | 0.8 | 253 | 202 | 29.7 | 86 | 2551 |
| 1M | 30 | 0.66 | 137 | 90 | 0.40 | 304 | 122 | 0.26 | 2000 | 521 | 2.6 | 386 | 1020 | 30.5 | 80 | 2437 |
| 1M | 100 | 1.31 | 204 | 267 | 0.55 | 400 | 220 | 0.52 | 2000 | 1035 | 6.2 | 185 | 1153 | 61.8 | 88 | 5441 |

# B  HYPER-PARAMETERS

Table 4 displays the values of the hyperparameters explored and a set of fixed parameters used throughout the model fitting process. The search spaces are carefully determined based on our thorough and extensive experiments. Search spaces for the hyperparameters of TreeNet are chosen based on their good default values discussed in the previous section. Note that the learning rate for TreeNet is larger because it has simpler structure than FCNN. This simpler structure does not require very small steps in gradient descent like FCNN does. Our experience shows that a learning rate of 0.01-0.02 works well. For all algorithms, we conduct a random search (with 20 trials) over the hyperparameters listed in the search space in Table 4. To prevent overfitting, we implement early stopping with a patience of 50 for all algorithms and $L_2$ regularization with $\lambda = 1$ for XGBoost. For FCNN and TreeNet, we use the ADAM optimizer with a learning rate decay of 0.995 per epoch. For TreeNet2, we fix number of cross layers at 2, number of bases at 5, dimension of projection layer at 20, learning rate at 0.02, batch size at 1% and select best number of epochs using early stopping. Since TreeNet2 is restricted to modeling up to three-way interactions, as a comparison, we also fitted

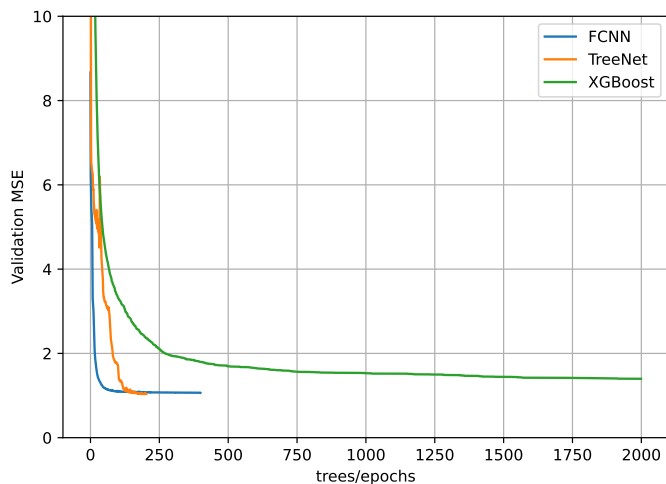

Figure 5: Validation loss curve for XGBoost, FCNN and TreeNet, for 1M and 100 variable case.

XGBoost3, where max depth is fixed at 3, learning rate at 0.05, subsample at 0.7, $L_2$ regularization at $\lambda = 1$ and select the best number of trees using early stopping.

Table 4: Hyper-parameter tuning settings.

| Algorithm | Search Space | Fixed Hyperparameters |
|---|---|---|
| TreeNet | <ul><li>Learning rate: [0.01, 0.02].</li><li>Batch size: [1%, 2%].</li><li>Number of cross layers: [0, 1, 2, 3].</li><li>Number of bases: [3, 5, 7].</li><li>Dimension of projection layer: [10, 20, 30, 40].</li></ul> | <ul><li>Decay = 0.995/epoch.</li><li>ADAM optimizer.</li><li>Early stopping with patience 50 and max number of epochs 400.</li></ul> |
| TreeNet2 | <ul><li>Learning rate: 0.02.</li><li>Batch size: 1%.</li><li>Number of cross layers: 2.</li><li>Number of bases: 5.</li><li>Dimension of projection layer: 20.</li></ul> | |
| FCNN | <ul><li>Layers: [20,10,5], [40,20,10], [60,30,15], [80,40,20], [100,50,25], [120,60,30,15], [100,50,25,12], [80,40,20,10], [10,20,40,20,10], [15,30,60,30,15].</li><li>Learning rate: [0.0001, 0.0005, 0.001, 0.002, 0.004, 0.008, 0.01, 0.015, 0.02].</li><li>Batch size: [1%, 2%, 4%].</li></ul> | |
| XGBoost | <ul><li>Max depth: [1, 2, 3, 4, 5, 6, 7].</li><li>Learning rate: [0.01, 0.02, 0.03, 0.04, 0.05, 0.06, 0.07].</li><li>Subsample: [0.5, 0.7, 1].</li></ul> | <ul><li>reg_lambda = 1 ($L_2$ regularization).</li><li>Early stopping with patience 50 and max number of trees 2000.</li></ul> |
| XGBoost3 | <ul><li>Max depth: 3.</li><li>Learning rate: 0.05.</li><li>Subsample: 0.7.</li></ul> | |
| GAMI-Net | <ul><li>Subnet nodes: [20, 40].</li><li>Subnet layers: [3, 5].</li><li>Interact num: [10, 20].</li><li>Reg clarity: [0.001, 0.01].</li><li>Batch size: [0.01, 0.025, 0.05, 0.1]</li><li>Learning rate: [0.0025, 0.005, 0.01].</li></ul> | <ul><li>Early stopping with patience 50 and max number of epoch 400.</li></ul> |
| TPNN2 | <ul><li>Num Ks: [20, 40].</li><li>Batch size: [0.01, 0.025, 0.05, 0.1]</li><li>Learning rate: [0.0001,0.0002, 0.0005, 0.001,0.002,0.005].</li></ul> | <ul><li>Early stopping with max number of epoch 400.</li></ul> |

## C  ABLATION STUDY

In this section, we study the sensitivity of TreeNet to its hyper-parameters, particularly the number of bases, learning rate and dimension of projection layer. We selected two real world datasets, HELOC and MSLR, for this study. HELOC is a small data with binary response and MSLR is a high-dimensional data with ordinal response (0 to 4). Using the default TreeNet2 settings, we adjust the three key hyper-parameters and evaluate model performance on test data. The parameter settings and model performances are given in 5.

As we can see, our default setting (bold face) may not be the best, but the difference with other settings is within the data splitting variation. For example, for HELOC data, default TreeNet2 has a testing AUC of 0.804, which is between the worse (0.802) and best (0.809), and the difference is smaller than the standard deviation caused by random data splitting (0.010, see 2). In addition, we can see small number of basis (3) suffices. This is consistent with the observations that feature effects tend to be simple (e.g., monotonic or very few turns) in real world data, hence does not require many spline bases. This is also made possible because our spline knots are not pre-determined but optimized, so even just one single basis can still be quite expressive. Finally, we see that for MSLR data which has 136 features, a small projection dimension like 10 still works well. This demonstrates the robustness of our algorithm.

Table 5: Parameter Ablation Study for TreeNet.

| LR | HELOC (AUC) | MSLR (MSE) | No. Basis | HELOC (AUC) | MSLR (MSE) | Proj. Dim | HELOC (AUC) | MSLR (MSE) |
|---|---|---|---|---|---|---|---|---|
| 0.0025 | 0.802 | 0.548 | 3 | 0.808 | 0.546 | 10 | 0.809 | 0.549 |
| 0.005 | 0.805 | 0.546 | 4 | 0.807 | 0.545 | **20** | **0.804** | **0.548** |
| 0.01 | 0.809 | 0.545 | **5** | **0.804** | **0.546** | 30 | 0.808 | 0.545 |
| **0.02** | **0.804** | **0.546** | 6 | 0.804 | 0.545 | 40 | 0.809 | 0.576 |
| 0.04 | 0.808 | 0.571 | 7 | 0.807 | 0.569 | 50 | 0.806 | 0.574 |

## D  COEFFICIENT EXTRACTION

To interpret the fitted XSN model, the first step is to expand the polynomial function that XSN represents. Recall that the spline layer creates $l$ bases, which is projected into $d$ reduced dimensions in the projection layer. Denote all the spline bases as $\boldsymbol{s}$, the projected variables as $\boldsymbol{h}$, $W^P$ and $b^P$ as weights and biases in the projection layer, we have $\boldsymbol{h} = W^P \boldsymbol{s} + b^P$. Going through the cross layers, we have $\boldsymbol{h}^{i+1} = \boldsymbol{h} \odot (W_i \boldsymbol{h}^i + b_i) + \boldsymbol{h}^i$, where $\boldsymbol{h}^0 = \boldsymbol{h}$. Each $\boldsymbol{h}^i$ is a $d$-dimensional vector where each element is a polynomial function of the spline bases $\boldsymbol{s}$. After $K$ cross layers, the output is $w^O \boldsymbol{h}^K + b^O$. Our goal is to extract the polynomial coefficients for the output neuron.

Starting from $\boldsymbol{h}^0 = W^P \boldsymbol{s} + b^P$. Each element $h_j^0$ is a simple linear function on $\boldsymbol{s}$, $j = 1, 2, ..., d$. If we store the intercepts and coefficients each into a tensor, we can represent $\boldsymbol{h}^0$ as a list, $[b^P, W^P]$, where $b^P$ is the $d \times 1$ vector of intercepts and $W^P$ is the $d \times l$ matrix of linear coefficients. More generally, each $\boldsymbol{h}^i$ is a vector of $d$ polynomial functions, each with max-order $i + 1$. The coefficients for order $k$ terms in each polynomial forms a $k$-dimensional tensor of size $(l, l, ..., l)$. Stacking all $d$ polynomial vertically, we can represent $\boldsymbol{h}^i$ as a sequence of tensors $[C_0^i, C_1^i, ..., C_{i+1}^i]$, where each $C_k^i$ contains $d$ stacked coefficient tensors of order $k$, and its size is $(d, l, l, ..., l)$.

Based on the relation $\boldsymbol{h}^{i+1} = \boldsymbol{h} \odot (W_i \boldsymbol{h}^i + b_i) + \boldsymbol{h}^i = (W^P \boldsymbol{s} + b^P) \odot (W_i \boldsymbol{h}^i + b_i) + \boldsymbol{h}^i$, we will derive an iterative updating algorithm for $C_k^{i+1}$ below.

Step 1: initialize $\boldsymbol{h}^{i+1} = \boldsymbol{h}^i$, which means $C_k^{i+1} = C_k^i$ for $k \le i + 1$ and $C_{i+2}^{i+1} = \boldsymbol{0}$.

Step 2: add the first term $b^P \odot b_i$, which updates $C_0^{i+1} = C_0^{i+1} + b^P \odot b_i$

Step 3: add the second term $(W^P \boldsymbol{s}) \odot b_i$, which updates $C_1^{i+1} = C_1^{i+1} + W^P \odot b_i$ where $b_i$ is broadcasted to each column of $W^P$.

Step 4: denote the new polynomial $W_i \boldsymbol{h}^i$ as a sequence of coefficient tensors $[D_0^i, D_1^i, ..., D_{i+1}^i]$, where $D_k^i = W_i C_k^i$ (note for $k \geq 2$, this becomes einsum along the second dimension of $W_i$ and first dimension of $C_k^i$).

Step 5: add the third term $b^P \odot (W_i \boldsymbol{h}^i)$, which updates $C_k^{i+1} = C_k^{i+1} + D_k^i \odot b^P$ for $0 \leq k \leq i+1$. Again $b^P$ is being broadcasted to each mode-1 fiber of $D_k^i$.

Step 6: add the last term $(W^P \boldsymbol{s}) \odot (W_i \boldsymbol{h}^i)$. Since this is an element-wise product, we can update $C_{k+1,j}^{i+1} = C_{k+1,j}^{i+1} + W_j^P \otimes D_{k,j}^i$, where $1 \leq j \leq d, 0 \leq k \leq i+1$, and $(u \otimes v)_{i,j_1,j_2,...,j_k} = u_i \times v_{j_1,j_2,...,j_k}$ is the outer product.

Given above updating formula, we can obtain the coefficient tensors after $K$ cross layers, denoted as $[C_0^K, C_1^K, ..., C_{K+1}^K]$. Then the coefficient tensor for output neuron is $[O_0, O_1, ..., O_{K+1}] = [b^O + w^O C_0^K, w^O C_1^K, ..., w^O C_{K+1}^K]$. The last step is to sum up coefficients for the same polynomial term. For example, $s_1 s_2$ and $s_2 s_1$ are the same second order term, so their corresponding coefficients $O_{2,12}$ and $O_{2,21}$ should be combined as one single final coefficient for $s_1 s_2$. More generally, for any tuple $(i_1, i_2, ..., i_k)$ where $i_1 \leq i_2... \leq i_k$, we sum up coefficients in $O_k$ for different permutations of $(i_1, i_2, ..., i_k)$. This gives us the final set of coefficients.

# E    TREENET INTERPRETATION TIME

In Section 6, we presented the model interpretation results. Here, we use 3way_cont as an example to show the time needed to perform such model interpretation. Recall that the data has 30 variables, and the fitted model is TreeNet2 with default settings (5 bases per variable and 2 cross layers). We consider three sample sizes, 10K, 100K, and 200K. The timing for each step is shown in Table 6.

As we mentioned in Section 6, after polynomial expansion, there are 585K polynomial coefficients, which is huge. By Holder's inequality, we can filter out most of the coefficients based on the L1-norm upper bound very quickly, with only around 20K terms left to calculate the L1-norm accurately. This makes filtering fast. The total filtering time is 15 seconds for 200K sample and less than 8 seconds for the 10K sample. In fact, since L1-norm is just mean absolute value, the results are very close. After filtering, we combine the terms under the same variable or variable set, and refit with the top $1, 2, 3, ...$ components. For this data, refitting stops at 14 components and the time for refitting is even less, just a few seconds. The top 14 components include two three-way interactions and six two-way interactions. Purification for each component takes an average of 19 seconds for 200K data and 8 seconds for 10K data. We ran purification for the same order interaction terms in parallel to speed up. The total purification time is 60 seconds for 200K data and 20 seconds for 10K data. The importance scores for purified effects are very close for 100K and 200K case (see Table 7). This is not surprising since each purification model is fitted with just two or three variables, therefore not a lot of observations are needed. This means for large sample sizes, we can use a subsample to perform the interpretation analysis.

| n | Filtering (s) | Refitting (s) | Purification (s/comp) | Purification (total) |
|---|---|---|---|---|
| 10K | 7.8 | 0.6 | 8 | 20 |
| 100K | 9.7 | 1.8 | 13.8 | 46 |
| 200K | 15.0 | 3.1 | 19.1 | 60 |

Table 6: Interpretation time for 3way_cont example

| | x1 | x3*x4 | x4*x5*x6 | x6 | x1*x2 | x5 | x4 | x2 | x3 | x1*x3 | x4*x5 | x1*x2*x3 | x5*x6 |
|---|---|---|---|---|---|---|---|---|---|---|---|---|---|
| 100K | 8.184 | 1.976 | 1.409 | 0.589 | 0.569 | 0.564 | 0.423 | 0.347 | 0.229 | 0.130 | 0.056 | 0.056 | 0.022 |
| 200K | 8.149 | 1.979 | 1.403 | 0.583 | 0.569 | 0.561 | 0.423 | 0.352 | 0.232 | 0.128 | 0.054 | 0.056 | 0.021 |

Table 7: Importance for purified effects for 100K and 200K data

## F  SIMULATION SETUP

The functional forms for main-effect only, two-way interaction and three-way interaction cases are as follows:

- Main effect only:
  - Main_cont: $f(\boldsymbol{x}) = x_1 + 2x_2^2 + 2(1 + x_3)^{1/3} + 2x_4(x_4 > 0) + \sin(2\pi x_5) + e^{x_6}$, where all main effects are continuous.
  - Main_jump: $f(\boldsymbol{x}) = x_1 + 2x_2^2 + 2(1 + x_3)^{1/3} + 2x_4(x_4 > 0) + \mathbb{I}(x_5 > 0) + 2\mathbb{I}(x_6 > 0.5)$, where $x_5, x_6$'s effects are discontinuous.

- Up to two-way interactions:
  - 2way_cont: $f(\boldsymbol{x}) = x_1 + 2x_2^2 + 2(1 + x_3)^{1/3} + 2x_4(x_4 > 0) + \sin(2\pi x_5) + e^{x_6} + 2x_1x_2 + 2\sin\left(\pi(x_3 + x_4)\right) + 2|x_5x_6|$, where all effects are continuous.
  - 2way_jump: $f(\boldsymbol{x}) = x_1 + 2x_2^2 + 2(1 + x_3)^{1/3} + 2x_4(x_4 > 0) + \mathbb{I}(x_5 > 0) + 2\mathbb{I}(x_6 > 0.5) + 2(x_1 > 0)(x_2 > 0) + 2(x_3 > 0)(x_4 > 0) + 2x_5(x_6 > 0)$, where the interactions are discontinuous.
  - 2way_pure: $f(\boldsymbol{x}) = 2x_1x_2 + \sin\left(\pi(x_3 + x_4)\right) + x_5\sin(\pi x_6)$, where all interactions are pure interactions (explained later).

- Up to three-way interactions:
  - 3way_cont: $f(\boldsymbol{x}) = x_1 + 2x_2^2 + 2(1 + x_3)^{1/3} + 2x_4\ (x_4 > 0) + \sin(2\pi x_5) + e^{x_6} + 2x_1x_2 + 2\sin\left(\pi(x_3 + x_4)\right) + 2|x_5x_6| + 3x_1e^{|x_2x_3|} + 3x_5\sin\left(\pi(x_4 + 1.5x_6)\right)$, where all effects are continuous.
  - 3way_jump: $f(\boldsymbol{x}) = x_1 + 2x_2^2 + 2(1 + x_3)^{1/3} + 2x_4(x_4 > 0) + \mathbb{I}(x_5 > 0) + 2\mathbb{I}(x_6 > 0.5) + 2(x_1 > 0)(x_2 > 0) + 2(x_3 > 0)(x_4 > 0) + 2x_5(x_6 > 0)$, where the interactions are discontinuous.
  - 3way_pure: $f(\boldsymbol{x}) = 2x_1x_2 + \sin\left(\pi(x_3 + x_4)\right) + x_5\sin(\pi x_6) + 2x_1x_2x_3 + 2x_4\sin\left(\pi(x_5 + x_6)\right)$, where all interactions are pure interactions (explained later).

For each case, we simulate 30 predictors $x_1, x_2 \ldots x_{30}$, following the independent Uniform$(-1, 1)$ distribution. Under such distribution, we have the conditional expectation $\mathbb{E}(x_1x_2 \mid x_1) = \mathbb{E}(x_1x_2 \mid x_2) = 0$. Similarly for $\sin(\pi(x_3 + x_4))$ and $x_5\sin(\pi x_6)$. Hence the three interaction terms in 2way_pure are pure interactions with no main effects. This is an uncommon case, but we use it to show the drawback of tree/tree-ensembles in modeling pure interactions. The response is simulated as $y = f(\boldsymbol{x}) + \epsilon, \epsilon \sim \mathcal{N}(0, 1)$. A total of 10,000 observations are simulated for each functional form, divided into 70% training and 30% validation data. A separate test set with 50,000 observations is used to assess the test performance. The test performances for all five algorithms and eight cases are shown in Table 8.

Table 8: Train MSE and Test MSE for five algorithms in eight simulation studies. Average and standard deviation (in parenthesis) over 10 trials are reported.

| Scenario | Train MSE | | | | | Test MSE | | | | |
|---|---|---|---|---|---|---|---|---|---|---|
| | TreeNet | FCNN | XGBoost | TreeNet2 | XGBoost3 | TreeNet | FCNN | XGBoost | TreeNet2 | XGBoost3 |
| Main_cont | 1.01 (0.02) | 1.25 (0.25) | 0.92 (0.04) | 1.00 (0.03) | 0.83 (0.03) | 1.02 (0.01) | 1.54 (0.02) | 1.04 (0.01) | 1.08 (0.02) | 1.06 (0.01) |
| Main_jump | 1.04 (0.02) | 1.09 (0.06) | 0.97 (0.02) | 1.05 (0.03) | 0.88 (0.03) | 1.04 (0.01) | 1.28 (0.04) | 1.02 (0.01) | 1.09 (0.01) | 1.04 (0.01) |
| 2way_cont | 0.99 (0.03) | 1.09 (0.11) | 0.37 (0.15) | 1.01 (0.06) | 1.05 (0.15) | 1.12 (0.02) | 1.45 (0.12) | 1.36 (0.03) | 1.22 (0.03) | 1.65 (0.08) |
| 2way_jump | 1.03 (0.04) | 1.36 (0.12) | 0.82 (0.08) | 1.10 (0.05) | 0.87 (0.05) | 1.19 (0.02) | 1.70 (0.09) | 1.12 (0.02) | 1.29 (0.02) | 1.13 (0.01) |
| 2way_pure | 0.96 (0.03) | 0.92 (0.14) | 0.21 (0.12) | 0.98 (0.06) | 0.55 (0.08) | 1.09 (0.02) | 1.36 (0.12) | 1.26 (0.02) | 1.19 (0.03) | 1.31 (0.03) |
| 3way_cont | 1.00 (0.09) | 1.51 (0.36) | 0.30 (0.19) | 1.02 (0.07) | 2.44 (0.23) | 1.36 (0.10) | 2.48 (0.54) | 2.90 (0.05) | 1.47 (0.05) | 3.58 (0.11) |
| 3way_jump | 1.14 (0.10) | 1.49 (0.16) | 0.64 (0.12) | 1.20 (0.06) | 0.96 (0.07) | 1.42 (0.04) | 2.07 (0.15) | 1.22 (0.02) | 1.50 (0.03) | 1.30 (0.02) |
| 3way_pure | 0.96 (0.04) | 0.98 (0.08) | 0.11 (0.09) | 0.99 (0.07) | 0.94 (0.20) | 1.22 (0.05) | 1.50 (0.10) | 1.79 (0.05) | 1.30 (0.04) | 2.09 (0.07) |

## G    REAL DATA DESCRIPTION

We looked at five public datasets in our real data study, briefly described below. Table 9 shows the size of the data and the number of features used in our model training. All non-binary features have been scaled to follow $\mathcal{N}(0, 1)$ distribution before model fitting.

- **Bike Sharing:** This is a public dataset hosted on the UCI machine learning repository. It contains hourly bike rental records from 2011 to 2012 in the Washington DC area. The goal is to predict the hourly bike rental counts. For count data, it is typical to apply the log transformation to stabilize the variance, so we model the log-count instead. The predictors include time information (hour, day of week, month, etc.) and weather information (temperature, humidity, wind speed, etc.).

- **California Housing**: This data was derived from the 1990 U.S. census, using one row per census block group. It includes detailed information about median income, population density, proximity to the coast, etc., to predict housing price. We scale the housing price by 100,000 so the MSE won't be huge.

- **MSLR**: This is the Microsoft Learning to Rank dataset. It consist of feature vectors extracted from query-url pairs along with relevance judgment labels from 0 (irrelevant) to 4 (perfectly relevant).

- **HELOC**: This data is a collection of anonymized credit applications, originally compiled by FICO. It includes various credit risk features from credit report data to predict if the applicant will 90 days past due or worse within the 24-month timeframe.

- **Spambase**: This is a public dataset hosted on the UCI machine learning repository. It is a collection of 4,601 email messages used for training and testing spam filtering algorithms. Each email is categorized as either "spam" or "not spam" (ham) based on 57 continuous attributes derived from the message content.

- **Magic**: This is a public dataset hosted on the UCI machine learning repository. The MAGIC Gamma dataset consists of simulated data from the MAGIC Gamma Telescope, used to classify gamma-ray events (signal) from cosmic rays (background) based on features describing the shape and characteristics of the Cherenkov light recorded by the telescope.

- **Madelon**: This is an artificial dataset, part of the NIPS 2003 feature selection challenge. This is a two-class classification problem with continuous input variables. The difficulty is that the problem is multivariate and highly non-linear.

| Dataset | Size | Features |
|---------|------|----------|
| Bike Sharing | 17379 | 11 |
| CalHousing | 20640 | 9 |
| MSLR | 1200192 | 136 |
| HELOC | 10459 | 23 |
| Spambase | 4601 | 57 |
| Magic | 19020 | 10 |
| Madelon | 2600 | 500 |

Table 9: Datasets

## H    FUNCTIONAL COMPONENTS FOR BIKE SHARING DATA

In this section, we provide additional details for the interpretability analysis of Bike sharing data. Fig. 6 shows correlation among the raw, unpurified functional components. As we can see, there are some hierarchical correlations among the interaction effect and its main effect, for example, *season*×*temp* and *temp* have a positive correlation of 0.6.

However, after purification, the hierarchical correlation reduces to nearly 0, as seen in Fig. 7. Note non-hierarchical correlation, like the correlation among main effects, or among two-way interactions, is a different issue.

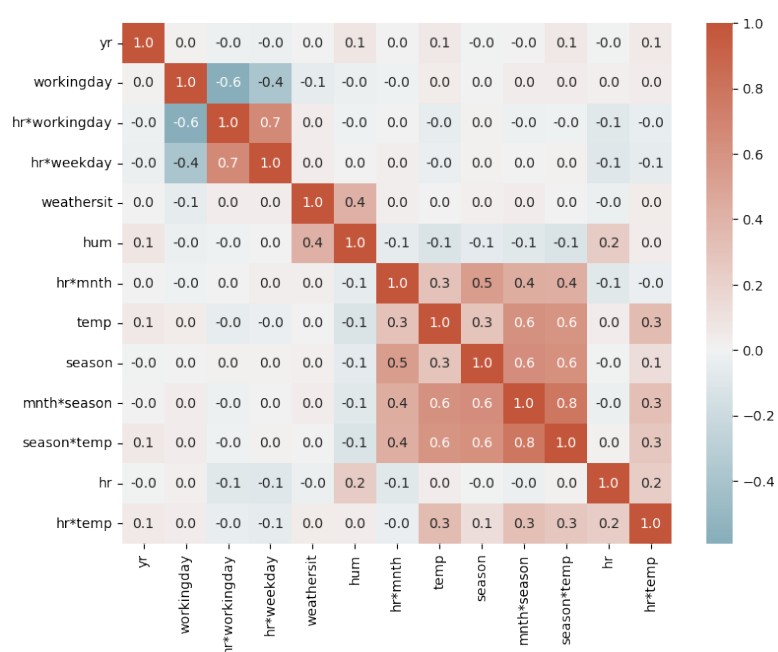

Figure 6: Correlation among raw functional components, before purification.

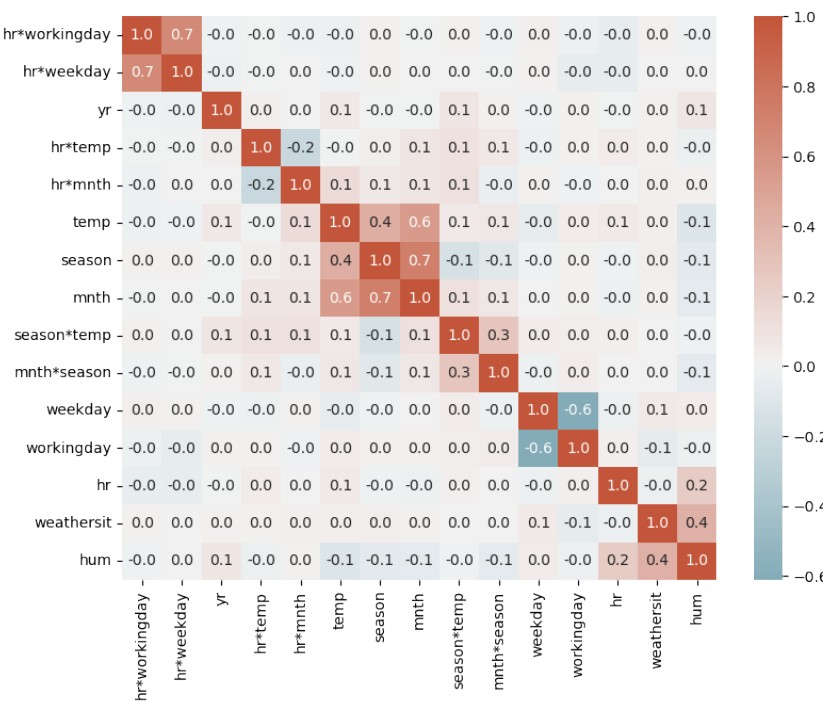

Figure 7: Correlation among purified functional components.

Finally, the importance for the purified effects should be re-calculated and is shown in Fig. 8. The main effect importance of *workingday* becomes close to 0 after purification.

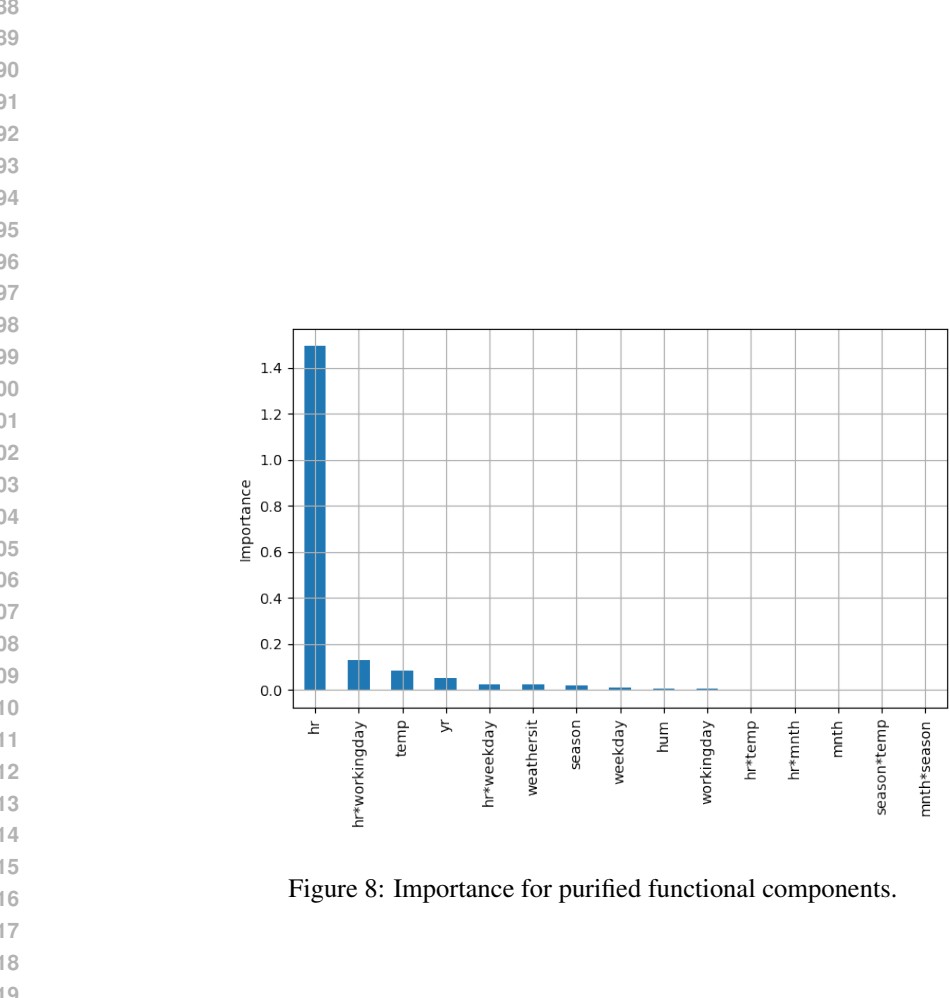

Figure 8: Importance for purified functional components.

