# OpenReview forum: "Cross Spline Net: A Simpler, More Interpretable and Unified Machine Learning Framework"
_ICLR.cc/2026/Conference — Submitted to ICLR 2026_

### Official Review · Reviewer_EPZP · 2025-10-15

**Soundness:** 2
**Presentation:** 1
**Contribution:** 1
**Rating:** 2
**Confidence:** 4

**Summary:**

This work introduces Cross Spline Net (XSN), a novel and interpretable machine learning framework. XSN features a simple yet effective architecture that unifies a range of existing ML techniques. Unlike traditional methods, it can leverage modern deep neural network (DNN) libraries for efficient optimization, while also offering built-in tools for interpretability.

**Strengths:**

- The explanations of the experimental settings are clear and sufficiently detailed.

**Weaknesses:**

- **Novelty:** The novelty of this work is questionable, as many similar approaches have been proposed in the past five years. For instance, [1, 2] also introduced unified frameworks bridging traditional ML methods and DNNs through polynomial formulations, while [3, 4] explored interpretable models based on polynomial neural networks and generalized additive models. Moreover, spline-based architectures for interpretability have already been utilized in KAN [5]. Although this work may differ in implementation details, the key distinctions are not clearly articulated in either the introduction or related work sections.
- **Claimed Advantages:** The authors argue that one advantage of XSN is the ability to apply DNN optimizers (e.g., Adam) to traditional ML techniques, whose original optimization procedures are often greedy, ad hoc, or not scalable to large datasets. However, many traditional ML algorithms involve convex optimization problems, for which stochastic optimizers may be unnecessary or even detrimental, potentially compromising desirable theoretical properties. Even if scalability is improved, such integration can typically be achieved without relying on XSN—for example, `scikit-learn` already supports stochastic gradient descent (SGD) for many estimators.
- **Framework Clarity:** Although the paper claims that XSN unifies a wide range of ML methods, the relationships between this framework and existing approaches are not clearly explained. Only a brief paragraph is dedicated to this topic, leaving the connections underdeveloped.
- **Experimental Results:** In the reported experiments, XSN underperforms several baseline methods in many cases. Moreover, training the TreeNet component appears significantly slower than both FCNNs and XGBoost. These results cast doubt on the practical utility and efficiency of the proposed framework.

[1] Zhang, Jiawei. "Rpn: Reconciled polynomial network towards unifying pgms, kernel svms, mlp and kan." _arXiv preprint arXiv:2407.04819_ (2024).

[2] Zhang, Jiawei. "RPN 2: On Interdependence Function Learning Towards Unifying and Advancing CNN, RNN, GNN, and Transformer." _arXiv preprint arXiv:2411.11162_ (2024).

[3] Duong, Viet, et al. "Cat: Interpretable concept-based taylor additive models." _Proceedings of the 30th ACM SIGKDD Conference on Knowledge Discovery and Data Mining_. 2024.

[4] Dubey, Abhimanyu, Filip Radenovic, and Dhruv Mahajan. "Scalable interpretability via polynomials." _Advances in neural information processing systems_ 35 (2022): 36748-36761.

[5] Liu, Ziming, et al. "Kan: Kolmogorov-arnold networks." _arXiv preprint arXiv:2404.19756_ (2024).

**Questions:**

- **Related Work:** The paper should include a dedicated section on related work, discussing prior studies on polynomial neural networks, existing unified ML frameworks, and Generalized Additive Models (GAMs). This would help position XSN within the broader research landscape and clarify its contributions relative to established methods.
- **Terminology Clarification:** The authors should clearly define the term _“spline transformation.”_ As presented, it does not appear to correspond to the conventional concept of spline interpolation or regression, which typically uses low-order piecewise polynomials to approximate functions. A precise explanation is necessary, as this term does not seem to be widely used or recognized in the existing literature.
- **Figure Quality:** All figures in the paper should be presented in vector format to ensure clarity and readability.

---

> ### Author Response · Authors · 2025-11-21
> **Author Response Part One**
>
> We thank the reviewer for the valuable feedback and additional references. We have made significant improvement in the updated paper to address the comments.
>
> **Novelty**: Thanks for the comment and references! We added a section of “Related Work” to clarify the novelties, please see the updated paper for details. Here, we briefly explain the novelty below (note for KAN [5], it applies a nonlinear transformation on the summation of all univariate splines, which is no longer a bounded-degree FANOVA model, so we are not comparing with KAN).
>
> First, we build higher-order FANOVA models in a scalable way, without the parameter or data explosion issue. We do so by using the efficient network design that cross-net provides. This is different from CAT [3] or SPAM [4] which uses low-rank approximation
> to approximate the high dimensional coefficient tensor. It is also different from GAMI-Net or TPNN which require an interaction filtering step first to avoid the parameter explosion issue.
>
> Second, we provide a comprehensive set of model simplification and interpretation tools after the model is fitted, filling an important gap. Without it, the interpretability goal won’t be achieved. Interpretation is challenging due to the large number of polynomial coefficients, but we provided efficient pruning and purification tools to tackle this problem. This is not done in [1], [2], [3] or [4].
>
> Finally, our unification framework is different from [1] and [2]. In [1] and [2], feature transformations are done in a pre-defined way in data expansion step and only coefficients are estimated. For example, Taylor expansion has a similar goal of fitting polynomial function, but it does so by creating all polynomial bases up to order d which grow exponentially. Our framework **engineers** all transformations and interactions, which is more efficient (the number of parameters only grows linearly), and can be useful in solving feature engineering problem, eg, knots selection problem.
>
> **Claimed Advantages**: As mentioned in the “Related Work” section, our bigger goal is to formulate as many problems as possible in a neural network framework and solve them using neural network. XSN is our first effort in this. Agree that some models are convex, like regression with L1/L2 regularization. However, it can only solve simple problems. A few examples here about why convex optimization is insufficient. (1) binning is a very useful technique in real-world modeling. However, how to find the best bin points are not trivial, and it is not a convex optimization problem. Heuristics like break-and-heal or tree splitting is used and it is done only one variable at a time. If we represent this binning model in neural network, with sharp-sigmoid function as activation, we can construct a NNET that presents binning process and the best bin points for all variables are found simultaneously with SGD. Regularizer can be easily added to control how many bin points are desired. (2) L1/L2 regularization is a convex approximation of L0 regularization. The downside is, it biases coefficients. There is a lot of work done to solve L0 regularization in an iterative way, where each iteration can still be convex. However, with NNET, we are no longer constrained by convexity and can plug-in any approximate L0 regularizer and solve it directly. (3) Convexity comes at a price. For example, for binning point selection, one can pre-define a set of eg, 100 candidates and do L1 selection in regression context, but this is not scalable with large data, and much worse when tensor products are considered. These (and XSN) are just a few examples here. The best part of NNET is, all one needs to do is to design a NNET architecture that represents the problem at hand, and you have a free, scalable and powerful solver.
>
> **Framework Clarity**: we explained the connection in the first two paragraphs of Section 3.1. We are unsure what’s missing. Can you please clarify?

---

> ### Author Response · Authors · 2025-11-21
> **Author Response Part Two**
>
> **Experimental Results**: We added GAMI-Net and second order TPNN in the updated Section 5, as well as two additional high-dimensional data (MSLR and Madelon). We can see TreeNet and TreeNet2 again show robust model performance with small overfitting gap. The performance is comparable with XGBoost on all datasets except for California housing and Madelon, where XGBoost is significantly better than **all other models**. For California housing, we suspect this is similar to the jumpy case in our simulation study, where the underlying data pattern is jumpy and more suitable for XGBoost model. For Madelon, this is a 'wide' data as it has many features (500) but only 2600 total observations (and training sample is even smaller after data splitting) and can overfit easily (see e.g., FCNN). TreeNet didn't do well initially (with test AUC around 0.63) since after spline transformation, the number of bases can exceed sample size and overfits. Therefore, we added an L0 regularizer to the projection layer (regularization is natively supported in neural networks), which penalizes the spline bases. This effectively mitigates the overfitting issue, and the model performance is much better (AUC 0.83). Note for all other datasets, regularization is not needed even for MSLR which has 136 features but sufficient sample size (1 million). Finally, comparing TreeNet with the other two interpretable models (GAMI-Net and TPNN2), TreeNet has **better performance for most cases**, except for Madelon where GAMI-Net is slightly better.
>
>
> | Data       | Metric     | TreeNet       | FCNN          | XGBoost       | TreeNet2      | XGBoost3      | GAMINet        | TPNN2          |
> |------------|-----------|---------------|---------------|---------------|---------------|---------------|---------------|---------------|
> | BikeShare  | Test MSE  | 0.114 (0.005) | 0.119 (0.003) | 0.104 (0.003) | 0.115 (0.004) | 0.110 (0.004) | 0.157 (0.041) | 0.190 (0.025) |
> | CalHousing | Test MSE  | 0.275 (0.012) | 0.272 (0.014) | 0.213 (0.007) | 0.276 (0.011) | 0.226 (0.007) | 0.323 (0.024) | 0.281 (0.011) |
> | MSLR       | Test MSE  | 0.543 (0.004) | 0.551 (0.001) | 0.530 (0.004) | 0.548 (0.006) | 0.548 (0.001) | 0.572 (0.005) | 0.561 (0.003) |
> | HELOC      | Test AUC  | 0.797 (0.010) | 0.790 (0.008) | 0.789 (0.009) | 0.796 (0.010) | 0.796 (0.008) | 0.799 (0.008) | 0.793 (0.009) |
> | Spambase   | Test AUC  | 0.981 (0.003) | 0.978 (0.003) | 0.986 (0.003) | 0.980 (0.003) | 0.985 (0.002) | 0.977 (0.003) | 0.974 (0.012) |
> | Magic      | Test AUC  | 0.935 (0.005) | 0.928 (0.005) | 0.933 (0.005) | 0.935 (0.004) | 0.933 (0.005) | 0.922 (0.007) | 0.907 (0.010) |
> | Madelon    | Test AUC  | 0.828 (0.022) | 0.628 (0.032) | 0.881 (0.047) | 0.819 (0.088) | 0.808 (0.022) | 0.840 (0.030) | 0.615 (0.01)  |
>
> **Regarding timing**, In the original run time table, we only reported the time per epoch/tree. In this revision, we added the total run time as this is the most relevant. FCNN and XSN take much fewer iterations to converge, since there can be 100 parameter updates in one epoch. For total run time, we can see the speed of XSN is comparable with FCNN and XGBoost for 100K case, and much faster than XGBoost in the 1M case since XGBoost requires many more iterations to converge (with a reasonable learning rate of 0.05 for XGBoost). Note that despite XSN creates multiple splines per variable, (1) the connection between spline layer and input layer is very sparse. One input variable is only connected to 5 basis, and the number of weight parameters for the spline layer is only 500 for 100 variable case (2) the projection layer further reduces the dimension to a small number. Finally, XSN can be run with GPU to be even faster.
>
> | |         | |TreeNet| | |FCNN| | |XGBoost| |
> |-------------------|-------|-------|-------|-------|-------|-------|-------|-------|-------|---|
>  |n | p | s/iter | iter | Total | s/iter | iter | Total | s/iter | iter | Total |
> | 100K | 30   | 0.38 | 104 | 40        | 0.25 | 204 | 51      | 0.018 | 1785 | 32 |
> | 100K | 100 | 0.60 | 159 | 96      | 0.26 | 192 | 51 |  0.049 | 1170 | 57 |
> | 1M | 30     | 0.66 | 137 | 90     | 0.40 | 304 | 122 |   0.26 | 2000 | 521 |
> | 1M | 100   | 1.31  | 204 | 267 |     0.55 | 400 | 220 |  0.52 | 2000 | 1035 |
>
> **Related Work**: Thanks for the comment! We have taken your advice and added a “Related Work” section.
>
> **Terminology Clarification**: Thanks for your comment. Here we use the term spline loosely to refer to any nonlinear transformation. We have clarified this in the paper.
>
> **Figure Quality**: Thanks for your comment. Will redo the plots if the paper can be accepted.

---

> > ### Comment · Reviewer_EPZP · 2025-11-21
> > **Reply for Rebuttal**
> >
> > 1. Thank you for adding the related work section. However, several of my concerns about novelty remain insufficiently addressed.
> >    - The construction of high-order polynomials is presented as a core contribution of this work, yet the manuscript appears to directly adopt CrossNet. The claim in Lines 92–93 is therefore not appropriate. In addition, the advantage of CrossNet over alternative approaches is still not clearly articulated.
> >    - The pruning method does not appear to be novel. Similar sparsification strategies—specifically, enhancing coefficient sparsity through iterative dropping and removal—were introduced in symbolic regression nearly a decade ago, for example in SINDy via the sequential thresholded least-squares algorithm.
> >
> > 2. Regarding the proposed framework, arguably the main contribution, the connections among its components should be explained more clearly. A summarizing table or schematic would greatly improve clarity.
> >
> > 3. Typo in the related work section: *SPAM*, not *SAPM*.

---

> ### Author Response · Authors · 2025-11-23
> **Author Response Part Three**
>
> > The construction of high-order polynomials is presented as a core contribution of this work, yet the manuscript appears to directly adopt CrossNet. The claim in Lines 92–93 is therefore not appropriate. In addition, the advantage of CrossNet over alternative approaches is still not clearly articulated.
>
> First, we have clearly mentioned cross-net’s strength (model interactions efficiently) and **weakness** that it can only model interactions of the simple form of x1x2 (line 150 of the paper). This weakness is mentioned in the original author’s paper (Wang et al. 2017, 2021) as well. Given the weakness, spline transformation is needed to capture interactions of more complicated form. That’s why spline layer is important. After spline transformation, the number of spline bases can grow to, eg, 5X, and that’s why a projection layer is important to reduce the dimension and make it computationally feasible, plus linear projection preserves the max order of the polynomial. Therefore, **all three building blocks, spline layer, projection layer and cross-net are indispensable to each other in our framework, not just cross-net**.
>
> Second, we want to re-emphasize that the modeling part is only **one** of the three core contributions. Our second contribution is we have a systematic and efficient way to interpret the model, which is equally important but not done in the literature. The interpretation is challenging due to the number of polynomial terms involved, which we will emphasize again in a later comment. Our third contribution is to use the unification power of XSN as an example to show the potential of using neural networks to solve modeling problems in an efficient and modern way.
>
> **Finally and most importantly, good, novel research does not mean to reinvent all the wheels, and this should be a common ground when assessing the novelty of the work.** For example, in the SPAM paper reviewer referenced, the authors used low-rank tensor decomposition which is a known mathematical theory, but it does not take away the novelty of the paper since it is a novel use of a math theory to solve a new problem. This is what we did with XSN as well, a novel use of spline, projection and cross net to solve a very challenging problem. **We believe the criterion for evaluating novelty should be based on: if the approach is new, if it works well (good performance and efficient), and if it is easy to understand and implement (so to attract usage in the real world). Our work checks all the boxes.**
>
> -	**Performance:** as our experiments have shown, our approach works better than XGBoost (which is the SOTA in tabular data modeling) for simulation data for non-jumpy cases (and much better in some cases), and is only worse than XGBoost in 2 out of 7 real world data, while being better or comparable on the others. Plus, it is better than TPNN and GAMI-Net in most cases except Madalon data where GAMI-Net is slightly better.
>
> -	**Efficiency:** we have only compared TreeNet with XGBoost and FCNN since we only compare with the fastest, and TreeNet has similar speed for 100K case (for 3way_cont data) and much faster than XGBoost for 1M case. To further show the efficiency, we added comparison with TPNN2 (learning rate 0.01, batch size 1%, number of K=10, number of interactions 20)  and GAMI-Net (learning rate 0.005, interaction number 20, subnets architecture [20,20,20]) below. Our method is much faster (order of magnitude faster than TPNN2). Plus, both TPNN2 and GAMI-Net only captures second order interaction, the model performance is much worse than TreeNet.
>
> | |         | |TreeNet| | |GAMI-Net| | |TPNN2| |
> |-------------------|-------|-------|-------|-------|-------|-------|-------|-------|-------|---|
>  |n | p | s/iter | iter | Total | s/iter | iter | Total | s/iter | iter | Total |
> | 100K | 30   | 0.38 | 104 | 40        | 0.7 | 212 | 148      | 13.3 | 140 | 1857 |
> | 100K | 100 | 0.60 | 159 | 96      | 0.8 | 253 | 202|  29.7| 86| 2551 |
> | 1M | 30     | 0.66 | 137 | 90     | 2.6 | 386 | 1020 |   30.5 | 80 | 2437 |
> | 1M | 100   | 1.31  | 204 | 267 |     6.2 | 185| 1153 |  61.8 | 88| 5441|
>
> -	**Ease to understand and implement:** Our work is obviously easy to understand and easy to implement (just stack up layers). This should be deemed as an advantage. The best framework should be one that’s easy to understand, easy to implement, yet effective. Our work achieves that.

---

> ### Author Response · Authors · 2025-11-23
> **Author Response Part Four**
>
> > The pruning method does not appear to be novel. Similar sparsification strategies—specifically, enhancing coefficient sparsity through iterative dropping and removal—were introduced in symbolic regression nearly a decade ago, for example in SINDy via the sequential thresholded least-squares algorithm.
>
> **It is very important to realize the scale and complexity of the problem to see the merits of our work**. As mentioned in the "Related Work" section, "interpretation is much more challenging since the number of coefficients and functional components can be huge". For model interpretation, what we have is a polynomial function that’s **expanded fully** (note that XSN can fit higher order without expanding the polynomial, but coming to interpretation, one still needs to expand the polynomial fully. This is still much better than having to fit all those terms in model training). As we mentioned in Section 6, there are 585,000 polynomial terms after expansion. That’s the biggest challenge here. Even populating all these 585K terms is challenging, since it takes $585K\times 10K\times 8$ Bytes, around 40GB to just store all of them, for a measly 10K sample size; and it will certainly take long CPU time to populate them as well. For 100K sample, the memory storage further grows to 400GB! There is not any least-square algorithms that can handle such scale **efficiently**. However, **our key novelty here is to solve this problem through Holder’s inequality.** As we explained in the paper, we calculate an upper bound of L1-norm first using Holder’s inequality. With this inequality, (1) we do not need to populate all 585K polynomial terms, just 500 spline bases (2) we calculate upper bound for each term in O(1) complexity instead of O(n). **The time and memory space savings are enormous**. In our paper, we used a very conservative L1-norm threshold of 0.001. The filtering only took 5 seconds with 12K terms left. If we slightly increase threshold to 0.02, there are only 1190 terms left, corresponding to 106 functional components. With only 106 components, any pruning method can be applied. **To conclude, the scale of the problem we are solving makes it challenging and we have an efficient solution for that.**

---

> ### Comment · Reviewer_EPZP · 2025-11-25
> **Final Reply for Rebuttal**
>
> Thank you for responding to my main concerns. I will raise my score to 4 to acknowledge the authors’ efforts. I still believe, however, that the writing could be strengthened, particularly the overall logic.

---

> > ### Author Response · Authors · 2025-11-25
> > **Thank you**
> >
> > Thank you for raising the score, we appreciate that! The writing can indeed be improved. Sometimes it is hard to see it from our own perspective about what's not clear. We will work on improving the writing in the next few days.

---

### Official Review · Reviewer_sxFF · 2025-10-17

**Soundness:** 2
**Presentation:** 3
**Contribution:** 2
**Rating:** 6
**Confidence:** 4

**Summary:**

This paper introduces Cross Spline Net (XSN), a novel machine learning framework that integrates spline transformations with cross-networks to create a unified, interpretable architecture for tabular data modeling. The authors demonstrate how XSN subsumes diverse model classes (including trees, MARS, and polynomial regression) under a single neural framework while leveraging modern optimizers like ADAM to overcome limitations of traditional greedy or ad-hoc training approaches.

**Strengths:**

The unification of spline transformations and cross-networks is an elegant contribution that bridges neural and non-neural approaches. By showing how XSN can approximate trees, MARS, and linear models, the authors provide a compelling framework for understanding these methods through a common lens. This conceptual clarity is rare and valuable.

The proposed interpretability tools (Algorithms 1–2) are methodologically sound and practically useful. The purification algorithm for enforcing hierarchical orthogonality is particularly noteworthy, as it addresses a key challenge in functional ANOVA decomposition. Visualizations of purified components (e.g., bike-sharing interactions) effectively showcase real-world insights.

**Weaknesses:**

The paper lacks theoretical guarantees regarding algorithmic convergence or the interplay among multiple splines, which is a significant concern for a novel framework. Without analysis of the optimization landscape—such as convexity properties or Lipschitz conditions—the reliability of XSN's training process remains uncertain. This omission is particularly noticeable given the hybrid nature of the architecture combining splines and cross-networks, where interactions between components could lead to complex loss surfaces. While empirical results are promising, theoretical grounding would substantially strengthen the methodological foundation.

The experimental evaluation, though extensive, has notable limitations in scope. High-dimensional scenarios (e.g., p > 100) are entirely absent, leaving scalability to modern large-scale datasets unverified. Given that spline transformations inherently expand feature space, performance degradation in high dimensions is a plausible concern that warrants investigation. Similarly, while runtime comparisons include 1M samples, the analysis focuses on computational speed rather than statistical performance at scale, leaving questions about stability and generalization in massive data regimes.

Baseline comparisons omit several critical competitors. The exclusion of modern interpretable methods like GAMI-Net and Explainable Boosting Machines is conspicuous, as these directly target the same niche of transparent yet performant tabular modeling. The comparison with XGBoost3 also appears methodologically skewed: constraining tree depth to 3 artificially limits its capacity to model higher-order interactions, while TreeNet2 explicitly permits up to three-way interactions. This asymmetry undermines the fairness of comparative claims.

Parameter sensitivity analysis is surprisingly absent for a framework advocating "off-the-shelf" usability. The robustness of TreeNet2 to variations in hyperparameters—particularly the number of spline bases (m), projection dimension (d), and learning rate—remains unexplored. This gap is problematic because real-world deployment often requires adaptation to diverse data characteristics, where sensitivity to default settings could limit practical utility. The paper's assertion about TreeNet2's generalizability would be more convincing with evidence of stability across hyperparameter perturbations.

Finally, the investigation of spline transformations is narrow, focusing exclusively on sigmoid-based functions. While suitable for approximating tree structures, this choice may not optimally capture other functional forms (e.g., periodic patterns or discontinuities). The framework's flexibility would be better demonstrated through a comparative analysis of alternative spline bases, which could reveal performance trade-offs and guide practical implementation.

I would be willing to raise my score if the above issues are well solved. Please correct me if I am wrong.

**Questions:**

Please refer to the weakness above.

---

> ### Author Response · Authors · 2025-11-21
> **Author Response Part One**
>
> We thank the reviewer for the positive feedback. We have made significant improvement in the updated paper to address the comments (a new "Related Work" section, additional interpretable models and higher-dimensional data, parameter ablation study, updated run time table, etc.)
>
> > theoretical guarantees regarding algorithmic convergence or the interplay among multiple splines
>
> The interplay between splines and cross-networks: Spline layer alone can approximate additive functions arbitrarily well (with enough bases). Cross-net is mainly used to capture interactions, which models product of neurons, eg, $x_1x_2$. This way of modeling interaction is better than the usual feedforward neural network (Wang et al., 2017; 2021). With spline layer, XSN becomes more expressive, eg, $\sigma(\alpha_{11}+\beta_{11}x_1)\sigma(\alpha_{21}+\beta_{21}x_2)$. For example, for an unknown interaction function f(x1, x2), with trained alpha and beta parameters, we can capture its effect with fewer terms and in better accuracy.
>
> For the loss function, it is complex and non-convex. To solve the polynomial regression problem in convex optimization, accounting for all interactions and high order terms, it requires one to populate all such polynomial terms, and only optimize the coefficients. This is infeasible for higher order models due to parameter explosion. As we mentioned in the added "Related Work" section, our work solves this problem in a scalable way. The splines are not pre-defined but trainable to be more flexible, and the interactions are engineered by layered cross-networks. This makes the problem non-convex, but the number of parameters grows only linearly; and there are advanced optimization routines (ADAM, ADAMw, etc) designed to solve hard, non-convex optimization problems in an efficient way. Proving the convergence of this highly complex function is difficult, but we will look into it in the future.
>
> > High-dimensional scenarios (e.g., p > 100) are entirely absent
>
> > Baseline comparisons omit several critical competitors.
>
> We added GAMI-Net and second order TPNN in the updated Section 5, as well as two additional high-dimensional data (MSLR and Madelon). We can see TreeNet and TreeNet2 again show robust model performance with small overfitting gap. The performance is comparable with XGBoost on all datasets except for California housing and Madelon, where XGBoost is significantly better than **all other models**. For California housing, we suspect this is similar to the jumpy case in our simulation study, where the underlying data pattern is jumpy and more suitable for XGBoost model. For Madelon, this is a 'wide' data as it has many features (500) but only 2600 total observations (and training sample is even smaller after data splitting) and can overfit easily (as we see in FCNN). TreeNet didn't do well initially (with test AUC around 0.63) since after spline transformation, the number of bases can exceed sample size and overfits. Therefore, we added an L0 regularizer to the projection layer (note regularization is supported natively in neural networks), which penalizes the (thousands of) spline bases. This effectively mitigates the overfitting issue, and the model performance is much better (0.83). Note for all other datasets, regularization is not needed even for MSLR which has 136 features but sufficient sample size (1 million). Finally, comparing TreeNet with the other two interpretable models (GAMI-Net and TPNN2), TreeNet has better performance for most cases, except for Madelon where GAMI-Net is slightly better.
>
>
> | Data       | Metric     | TreeNet       | FCNN          | XGBoost       | TreeNet2      | XGBoost3      | GAMINet        | TPNN2          |
> |------------|-----------|---------------|---------------|---------------|---------------|---------------|---------------|---------------|
> | BikeShare  | Test MSE  | 0.114 (0.005) | 0.119 (0.003) | 0.104 (0.003) | 0.115 (0.004) | 0.110 (0.004) | 0.157 (0.041) | 0.190 (0.025) |
> | CalHousing | Test MSE  | 0.275 (0.012) | 0.272 (0.014) | 0.213 (0.007) | 0.276 (0.011) | 0.226 (0.007) | 0.323 (0.024) | 0.281 (0.011) |
> | MSLR       | Test MSE  | 0.543 (0.004) | 0.551 (0.001) | 0.530 (0.004) | 0.548 (0.006) | 0.548 (0.001) | 0.572 (0.005) | 0.561 (0.003) |
> | HELOC      | Test AUC  | 0.797 (0.010) | 0.790 (0.008) | 0.789 (0.009) | 0.796 (0.010) | 0.796 (0.008) | 0.799 (0.008) | 0.793 (0.009) |
> | Spambase   | Test AUC  | 0.981 (0.003) | 0.978 (0.003) | 0.986 (0.003) | 0.980 (0.003) | 0.985 (0.002) | 0.977 (0.003) | 0.974 (0.012) |
> | Magic      | Test AUC  | 0.935 (0.005) | 0.928 (0.005) | 0.933 (0.005) | 0.935 (0.004) | 0.933 (0.005) | 0.922 (0.007) | 0.907 (0.010) |
> | Madelon    | Test AUC  | 0.828 (0.022) | 0.628 (0.032) | 0.881 (0.047) | 0.819 (0.088) | 0.808 (0.022) | 0.840 (0.030) | 0.615 (0.01)  |

---

> ### Author Response · Authors · 2025-11-21
> **Author Response Part Two**
>
> > XGBoost3 comparison to TreeNet2
>
> XGBoost3 is compared to TreeNet2 since both are restricted to up to three-way interaction (which is already one order higher than, eg, EBM or GAMI-Net and sufficient for many datasets, see Lou et al. (2013) and Dubey et al., (2022)). With deeper depth (>=4), XGBoost can model 4th order interaction, which is unfair to compare against TreeNet2. Nevertheless, we also have the unconstrained XGBoost, which is compared against the unconstrained TreeNet.
>
> > Parameter sensitivity analysis
>
> We added an ablation study in Appendix, where we adjust the values for 3 key HPs (learning rate, number of bases per variable and projection dimension). TreeNet is robust to HPs, with performance difference within data splitting variation on two representative real-world datasets (MSLR and HELOC). For example, for HELOC data, default TreeNet2 (bold face) has a testing AUC of 0.804, which is between the worse (0.802) and best (0.809), and the difference is smaller than the standard deviation caused by random data splitting (0.010, see Table 2 in the paper). In addition, we can see small number of basis per variable (3) suffices. This is consistent with the observations that feature effects tend to be simple (e.g., monotonic or very few turns) in real world data, hence does not require many spline bases. This is also made possible because our spline knots are not pre-determined but optimized, so even just one single basis can still be quite expressive. Finally, we see that for MSLR data which has 136 features, a small projection dimension like 10 still works well. This demonstrates the robustness of our algorithm.
>
> | LR      | HELOC (AUC) | MSLR (MSE) | No. Basis | HELOC (AUC) | MSLR (MSE) | Proj. Dim | HELOC (AUC) | MSLR (MSE) |
> |---------|-------------|------------|-----------|-------------|------------|-----------|-------------|------------|
> | 0.0025  | 0.802       | 0.548      | 3         | 0.808       | 0.546      | 10        | 0.809       | 0.549      |
> | 0.005   | 0.805       | 0.546      | 4         | 0.807       | 0.545      | **20**    | **0.804**   | **0.548**  |
> | 0.01    | 0.809       | 0.545      | **5**     | **0.804**   | **0.546**  | 30        | 0.808       | 0.545      |
> | **0.02**| **0.804**   | **0.546**  | 6         | 0.804       | 0.545      | 40        | 0.809       | 0.576      |
> | 0.04    | 0.808       | 0.571      | 7         | 0.807       | 0.569      | 50        | 0.806       | 0.574      |
>
> > investigation of spline transformations
>
> We have implemented other transformations, including RELU (piece-wise linear transformation) and Sharp-Sigmoid (with a fixed large beta but trainable alpha in $\sigma(\alpha+\beta x)$ to approximate indicator function) which are very popular in real-world applications. For XSN with RELU transformation, the model performance is similar to TreeNet, but it extrapolates linearly instead of flattening out. So the out of distribution performance is different (sigmoid can be more robust). For sharp-sigmoid, the model performance is not as good. This is because it will need many bases to approximate a continuous function. However, sharp-sigmoid is useful when optimal binning is considered, where a small set of best bin-points needs to be selected. This avoids the issue of ad-hoc optimization methods (like break-and-heal), another example of how neural network can provide a more modern and efficient solution to old problems.

---

### Official Review · Reviewer_o2Ft · 2025-10-29

**Soundness:** 2
**Presentation:** 2
**Contribution:** 1
**Rating:** 2
**Confidence:** 3

**Summary:**

They proposed a new machine learning framework called Cross Spline Net (XSN), which includes TreeNet - an interpretable model that estimates each feature’s main effect using sigmoid-based basis functions, applies a linear dimension-reduction layer, and captures feature interactions through a cross layer. The model was evaluated on real-world datasets.

**Strengths:**

**(S1).** The proposed model TreeNet seems to effectively capture higher-order feature interactions.

**Weaknesses:**

**(W1).** The literature review appears to be somewhat limited.
Since the authors propose TreeNet, a model that can be interpreted from the perspective of a functional ANOVA model, the literature review should also include relevant studies related to functional ANOVA model.
That is, I believe it would be necessary to discuss recent interpretable models based on the functional ANOVA model, such as NAM ([1]), NBM ([2]), and ANOVA-TPNN ([3]).
In particular, ANOVA-TPNN estimates functional ANOVA components using a tensor product of **sigmoid**-based basis functions, so a comparison with this model would be important.


**(W2).** The baseline models used in the experiments seem to be insufficient, and it would be beneficial to include an additional comparison with functional ANOVA–based models such as NAM ([1]), NBM ([2]), and ANOVA-TPNN ([3]).

**(W3).** The real datasets used in the experiments may not have sufficiently large sample sizes or input feature dimensions.
I think the proposed model requires additional evaluation on larger real-world datasets, such as those used in [2] or [3].

**(W4).** It is difficult to understand how the components of the functional ANOVA model are obtained from a trained TreeNet in Section 2.3.
A more detailed explanation of this part seems necessary, particularly including Algorithm 1.
For example, in line 183 it says “l bases” what exactly does "l" refer to? Does it represent the dimension of the features after dimension reduction?

**(W5).**
In the paper, it is not clear what aspects of the proposed model are novel compared to existing models.
In my view, the proposed model and methodology seem to lack sufficient novelty to make a substantial contribution.
It would be helpful if the authors could better highlight which aspects of the proposed model and methodology are novel compared to existing approaches.



**References**

[1]. Agarwal, Rishabh, et al. "Neural additive models: Interpretable machine learning with neural nets." Advances in neural information processing systems 34 (2021): 4699-4711.

[2]. Radenovic, Filip, Abhimanyu Dubey, and Dhruv Mahajan. "Neural basis models for interpretability." Advances in Neural Information Processing Systems 35 (2022): 8414-8426.

[3]. Park, Seokhun, et al. "Tensor Product Neural Networks for Functional ANOVA Model." International conference on machine learning  (2025).

**Questions:**

**(Q1).**
According to the experimental results in Section A, the run time for model training does not appear to increase significantly with respect to n and p.
However, in the case of Algorithm 2, the run time is expected to increase rapidly as p grows.
Therefore, I think it would be necessary to include additional experiments to verify whether the proposed TreeNet model can still provide interpretable results in a computationally feasible manner when p becomes large.
Specifically, would it be possible to provide experiments that analyze how the run time of Algorithm 2 changes as p increases?


**(Q2).**
If, as stated in **(W4)**, "l bases" indeed refers to the reduced feature dimension, then the components obtained through the purifying algorithm appear to be based on dimension-reduced latent features rather than the original input features.
Could the authors please clarify how TreeNet can still be regarded as an interpretable model under this setting?

**(Q4).**
Is it correct that the number of trees for the XGB model was set to 100 in the experiments? Fixing it at 100 seems rather small. Could you please explain the reason for this choice?

**[Minor]**

**(Q4).**
In Table 6, the input feature dimension of the Calhousing dataset is listed as 9, but isn’t it actually 8?

---

> ### Author Response · Authors · 2025-11-21
> **Author Response Part One**
>
> We thank the reviewer for the valuable feedback and additional references. We have made significant improvement in the updated paper to address the comments.
>
> **(W1)** Good point! We added a section of “Related Work” to clarify our contributions. Our work solves two key issues in functional-ANOVA (FANOVA) models. First, our method builds higher-order FANOVA models in a scalable way, without the parameter or data explosion issue. Second, we provide a comprehensive set of model simplification and interpretation tools to understand the fitted model, which fills an important gap especially for higher-order models (without it, the interpretability benefit is not achieved). Finally, through our unification effort, we want to promote solving as many problems as possible by neural network (with splines and crosses), in a modern and efficient way. In fact, we are working on some new directions that uses XSN to create transparent feature engineering (not black-box feature engineering that black-box ML does).
>
> **(W2)** Agree! We added GAMI-Net and TPNN2 (second order tensor product neural network) to our real data study and also included two additional real datasets with high dimension (MSLR and Madelon). We can see TreeNet and TreeNet2 again show robust model performance with small overfitting gap. The performance is comparable with XGBoost on all datasets except for California housing and Madelon, where XGBoost is significantly better than all other models. For California housing, we suspect this is similar to the jumpy case in our simulation study, where the underlying data pattern is jumpy and more suitable for XGBoost model. For Madelon, this is a 'wide' data as it has many features (500) but only 2600 observations (training sample is even smaller) and can overfit easily (as we see for FCNN). TreeNet didn't do well initially (with test AUC around 0.63) since after spline transformation, the number of bases can exceed sample size and overfits. Therefore, we added an L0 regularizer to the projection layer (regularization is supported natively in neural network), which penalizes the spline bases. This effectively mitigates the overfitting issue, and the model performance is much better (0.83 AUC). Note for all other datasets, regularization is not needed even for MSLR which has 136 features but large sample size (1 million). Finally, comparing TreeNet with the other two interpretable models (GAMI-Net and TPNN2), TreeNet has better performance for most cases, except for Madelon where GAMI-Net is slightly better.
>
>
> | Data       | Metric     | TreeNet       | FCNN          | XGBoost       | TreeNet2      | XGBoost3      | GAMINet        | TPNN2          |
> |------------|-----------|---------------|---------------|---------------|---------------|---------------|---------------|---------------|
> | BikeShare  | Test MSE  | 0.114 (0.005) | 0.119 (0.003) | 0.104 (0.003) | 0.115 (0.004) | 0.110 (0.004) | 0.157 (0.041) | 0.190 (0.025) |
> | CalHousing | Test MSE  | 0.275 (0.012) | 0.272 (0.014) | 0.213 (0.007) | 0.276 (0.011) | 0.226 (0.007) | 0.323 (0.024) | 0.281 (0.011) |
> | MSLR       | Test MSE  | 0.543 (0.004) | 0.551 (0.001) | 0.530 (0.004) | 0.548 (0.006) | 0.548 (0.001) | 0.572 (0.005) | 0.561 (0.003) |
> | HELOC      | Test AUC  | 0.797 (0.010) | 0.790 (0.008) | 0.789 (0.009) | 0.796 (0.010) | 0.796 (0.008) | 0.799 (0.008) | 0.793 (0.009) |
> | Spambase   | Test AUC  | 0.981 (0.003) | 0.978 (0.003) | 0.986 (0.003) | 0.980 (0.003) | 0.985 (0.002) | 0.977 (0.003) | 0.974 (0.012) |
> | Magic      | Test AUC  | 0.935 (0.005) | 0.928 (0.005) | 0.933 (0.005) | 0.935 (0.004) | 0.933 (0.005) | 0.922 (0.007) | 0.907 (0.010) |
> | Madelon    | Test AUC  | 0.828 (0.022) | 0.628 (0.032) | 0.881 (0.047) | 0.819 (0.088) | 0.808 (0.022) | 0.840 (0.030) | 0.615 (0.01)  |
>
> **(W3)** Please see our response to W2.
>
> **(W4)** Sorry for the confusion. We added an example in Section 3.3 to clarify. For example, assume $\gamma_{1}\sigma(\alpha_{11} + \beta_{11} x_1), \quad
> \gamma_{2}\sigma(\alpha_{12} + \beta_{12} x_1),  \quad
> \gamma_{3}\sigma(\alpha_{11} + \beta_{11} x_1)\times \sigma(\alpha_{12} + \beta_{12} x_1), \quad \gamma_{4}\sigma(\alpha_{21} + \beta_{21} x_2), \quad \gamma_{5}\sigma(\alpha_{11} + \beta_{11} x_1)\times \sigma(\alpha_{12} + \beta_{12} x_2)$ are the remaining polynomial terms after filtering. The first three terms all capture part of the main-effect of $x_1$, the fourth term captures the main-effect of $x_2$ and the last term captures the interaction-effect of $x_1, x_2$. The component $f_1(x_1)$ is simply the sum of the first three terms.
>
> For "l bases", “l” is the total number of distinct bases after spline transformation but before the projection/dimension reduction layer. We also clarified this in the paper.
>
> **(W5)** Please see our response to W1.

---

> > ### Author Response · Authors · 2025-11-21
> > **Author Response Part Two**
> >
> > **(Q1)** For Algorithm 2 (Purification), the remaining functional components after pruning is more important than p. When p increases, the number of important main-effects and interactions do not necessarily grow much. We assume that a relatively small number of components will explain most of the signal in the data. This is a reasonable assumption that people make. Otherwise, if there are many equally important but small effects, interpretation won’t be possible as there are too many small things to interpret. This is also referred to as the “bet on sparsity” principle (see https://www2.stat.duke.edu/~banks/218-lectures.dir/dmlect9.pdf). I hope this answers your question. If you are interested in run time of the interpretation tools, please refer to our response to reviewer XWEf.
> >
> > **(Q2)** Please see our response to W4.
> >
> > **(Q3)** Thanks for catching that! We forgot to add the max number of epochs/trees for the training. We did rigorous tuning for all the models and picked sufficiently large numbers for the max trees/epochs. The max number of epochs for FCNN/TreeNet is up to 400, and the max number of trees for XGBoost is up to 2000 (two thousand). We only used a small value of 100 in the run time analysis. We clarified this in the revision.
> >
> > **(Q4)** These are the 9 variables in our data: 'longitude', 'latitude', 'housing_median_age', 'total_rooms', 'total_bedrooms', 'population', 'households', 'median_income', 'ocean_proximity'.

---

> > > ### Comment · Reviewer_o2Ft · 2025-11-25
> > >
> > > Thank you for the authors’ thoughtful response.
> > > However, I still have a few remaining questions. They are as follows:
> > >
> > > **Question 1.**
> > >
> > > Is it correct to say that the key contribution of the paper is using bases and estimating higher-order interactions through the cross-layer, thereby improving scalability compared to existing fANOVA models?
> > >
> > > Neural Basis Model (NBM, [1]) introduces the notion of bases to improve the scalability of NAM, enabling the estimation of higher-order interactions even in high-dimensional datasets.
> > > ANOVA-TPNN ([2]) also adopts the basis concept from NBM to propose the NBM-TPNN, which can estimate interactions based on these bases—without even requiring purification.
> > > Given this, in what ways does TreeNet offer advantages over these models?
> > >
> > >
> > > **Question 2.**
> > >  As far as I know, even though unnecessary components are removed, the computational cost of post-hoc identification (purification) is still $O(n^{d})$, where $d$ is the maximum order of selected component and $n$  is the size of dataset.
> > > Therefore, I have concerns about whether interpretation can be provided in practice for large-scale datasets such as Microsoft or Yahoo, where both $n$ and $p$ are large.
> > >  That is, I am curious about the runtime required to obtain interpretations from TreeNet—including the purification procedure—on large-scale datasets such as Microsoft.
> > >
> > > Also, I would like to ask whether the experiment in the rebuttal of XWEf corresponds to the one shown in Appendix A of the revised paper.
> > > In that section, it seems that only the training time of TreeNet is reported. Is it right?
> > >
> > > In addition, the number of cross-layer blocks in TreeNet is fixed to 2 in the current runtime experiments.
> > > It would be important to report results with larger numbers of cross-layer blocks as well.
> > >
> > > $\newline$
> > > $\newline$
> > >
> > > **Question 3.**
> > > It would be helpful if the authors could clarify where the components used for fANOVA-based interpretation are produced.
> > > In particular, in the revised paper (lines 264–265), could you explain where $\gamma$ is defined ?
> > > Additionally, are the terms $\gamma_{1}\sigma(\alpha_{1}+\beta_{1}x_{1}),...$ obtained from the outputs after passing through the cross-layer ?
> > > A clear and specific expression is needed to show exactly how these terms are derived.
> > >
> > > $\newline$
> > > $\newline$
> > >
> > > **Question 4.**
> > > It appears that TreeNet’s prediction performance is sometimes significantly worse than XGB, even though TreeNet is designed to estimate higher-order interactions (e.g., in the calhousing and madelon datasets).
> > > Could the authors clarify the reason for this performance gap?
> > >
> > > $\newline$
> > > $\newline$
> > >
> > > **Question 5.**
> > > The Calhousing dataset is known to contain 8 features according to the sklearn repository ([3]). Is the Calhousing dataset used in your experiments different from the one provided by sklearn?
> > >
> > > $\newline$
> > > $\newline$
> > >
> > > **Question 6.**
> > > It seems that references for the real datasets used in the experiments are not provided. Is this correct?
> > >
> > >
> > > $\newline$
> > > $\newline$
> > >
> > > **References**
> > >
> > > [1]. Radenovic, Filip, Abhimanyu Dubey, and Dhruv Mahajan. "Neural basis models for interpretability." Advances in Neural Information Processing Systems 35 (2022): 8414-8426.
> > >
> > > [2]. Park, Seokhun, et al. "Tensor Product Neural Networks for Functional ANOVA Model." arXiv preprint arXiv:2502.15215 (2025).
> > >
> > > [3]. https://scikit-learn.org/stable/modules/generated/sklearn.datasets.fetch_california_housing.html

---

> ### Author Response · Authors · 2025-11-29
> **Author Response Part Three**
>
> We thank the reviewer for these questions. These are important questions to address to understand the merits and contributions of our work.
>
> **Question 1**: The core issue of FANOVA modeling and interpretation is scalability given the exploding number of parameters and data. Our two key contributions are therefore addressing modeling and interpretation in a scalable way. For interpretation part, we will answer in question #2. For modeling, our framework adopts a simple yet effective design integrating spline layer, projection layer and cross layer. Cross layer is efficient in modeling interactions, but it only captures interactions of the simple form of x1x2. Given this limitation, spline transformation is necessary to capture interactions of more complicated form. Finally, projection layer is important to reduce the dimension after spline transformation and make it computationally feasible, plus linear projection preserves the max order of the polynomial. This is how XSN addresses the modeling part in a scalable way without parameter or data explosion.
>
> **Comparing with NBM**: NBM does not solve the parameter or data explosion issue. For example, NB2M extends NBM with interaction modeled as $f_{ij}(x_i,x_j)=\sum_{k=1}^B u_k(x_i,x_j) b_{ijk}$. The B basis functions are shared for all variable pairs to reduce the parameters for basis functions, but the coefficients $b_{ijk}$ still grow exponentially (as order increases) when all possible interactions are considered ($O(p^2 B)$). Similarly, the number of spline bases $u_k(x_i,x_j)$ also grows exponentially. Compared with our approach, we don’t have data or parameter explosion issue, and we don’t need to rely on regularization to control overfitting as much as NBM.
>
> **Comparing with ANOVA-TPNN**: TPNN has the same parameter/data explosion issue if all interactions are considered, but it avoids this core issue by requiring an interaction filtering algorithm to reduce the dimensionality first. Under such simplification, the problem is much easier and there are many ways to model the interactions (GAMI-Net and EBM can all be easily extended to higher dimension). However, high-order interaction detection is very challenging. The “FAST” interaction filtering algorithm, developed by EBM and used by GAMI-Net as well, only works for 2nd order. FAST exhaustively searches for all variable pairs and for each pair it fits a crude 4-quadrant model with optimized cut-point on each variable. To generalize to any order $d$, the complexity will be $O(K^d p^d)$ where $K$ is the number of candidate cut points for each feature and $p$ is number of features. This is infeasible. Our method does not require the filtering step and we solve the problem head-on. In fact, our method can be simplified and **turned into a general-purpose interaction filtering algorithm**, since filtering only requires crude approximation, not full fit. We can simply use one sigmoid basis per variable and our TreeNet will fit all pairs with optimized spline transformation in one go. This makes the impact of our work even greater.
>
> Finally, compared with TPNN2, our approach has better performance (see Table 2) and much faster speed. In the table below, we compare the runtime of TreeNet with GAMI-Net and TPNN2. **Our method is much faster** due to our efficient architecture design.
>
> | |         | |TreeNet| | |GAMI-Net| | |TPNN2| |
> |-------------------|-------|-------|-------|-------|-------|-------|-------|-------|-------|---|
>  |n | p | s/iter | iter | Total | s/iter | iter | Total | s/iter | iter | Total |
> | 100K | 30   | 0.38 | 104 | 40        | 0.7 | 212 | 148      | 13.3 | 140 | 1857 |
> | 100K | 100 | 0.60 | 159 | 96      | 0.8 | 253 | 202|  29.7| 86| 2551 |
> | 1M | 30     | 0.66 | 137 | 90     | 2.6 | 386 | 1020 |   30.5 | 80 | 2437 |
> | 1M | 100   | 1.31  | 204 | 267 |     6.2 | 185| 1153 |  61.8 | 88| 5441|

---

> > ### Author Response · Authors · 2025-11-29
> > **Author Response Part Four**
> >
> > **Question 2**: Interpretation is challenging due to the huge number of terms and components after polynomial expansion, and that’s the gap we are trying to fill. To answer your question, we first explain our approach and then use MSLR as an example.
> >
> > Our approach involves pruning and purification. Given the size of the problem, pruning is actually the most important part (in the future, we are exploring adding L1 regularization to reduce dimension of the coefficient table). Calculating L1-norms can be very expensive when the number of terms is huge. We solve this issue using Holder’s inequality, which reduces the computation for L1 norm to O(1) instead of O(n). For the remaining terms after filtering, further pruning by functional components becomes much easier. After pruning, we assume only limited number of components are important and kept. This is a reasonable assumption that other models (GAMI-Net, TPNN) also assume. To purify a component $f_S(x_S)$ of order $d$, it involves fitting a model with up to $d-1$ order interactions. This can be done by fitting an xgboost with max_depth $d-1$, or fitting a TreeNet like we did. It is a simple model fitting problem with only $d$ variables ($d$ is small). The complexity is linear in $n$ and $d$, not $n^d$. For large n, a subsample can be used when calculating L1-norm (since it is simple average), or when purifying each component since $d$ is small and there is no overfitting concern ($f_S(x_S)$ has no noise).
> >
> > Using MSLR as an example, for our default TreeNet2 model with 5 bases and 2 cross layers, total number of splines is over 600. Total number of coefficients for fully expanded polynomial is 48.4 Million! Use 100K subsample in interpretability analysis. Filtering using Holder’s inequality took 342 seconds, with only 26.9K terms left (using default threshold 0.001). Accurately calculating the L1-norms for the 26.9K terms took 7 seconds and there are only 8K terms left. Pruning based on functional components again took just 6 seconds and the selected top 28 components yields a test MSE of 0.592, only 8.4% higher than the original TreeNet2 model (0.547). In the top 28 components, there are 9 two-way interactions and no three-way interactions. Purification for all two-way interactions is done in parallel and it took 43 seconds in total. With 200K samples, the purification runtime increases about 50% but the results are quite stable.
> >
> > > In that section, it seems that only the training time of TreeNet is reported. Is it right?
> >
> > Yes. We added the run time for interpretation in Appendix E of the updated paper.
> >
> > > The number of cross-layer blocks in TreeNet is fixed to 2
> >
> > Just to clarify, TreeNet2 fixed cross-layer blocks to 2, while TreeNet tunes the number of cross layers.
> >
> > **Question 3**: Thanks for the question. We skipped the polynomial expansion part due to the page size limit but we have added it in Appendix D. Briefly speaking, we derive an iterative coefficient updating formula based on the relationship $ x^{i+1}=x \odot (W_i x^i+b_i) +x^i $. Please see Appendix D for details.
> >
> > **Question 4**: For calhousing data, longitude and latitude are the two most important variables (based on permutation based importance). However, their effects are jumpy (TPNN paper has some effect plots showing the jumpiness, eg, Figure 1). As we have seen in the simulated data case, TreeNet performs worse than xgboost for jumpy case. This is a challenge for all neural network based methods (FCNN, TPNN, GAMI-Net, etc.) since they are typically smooth.
> >
> > For madelon data, it is an artificially created data with 20 real features and 480 random features. This data can cause overfit easily, as we can see for FCNN, TPNN2 and TreeNet without regularization. XGBoost did well on this data, because during each tree split, it selects the most predictive variable to split. This makes it less affected by the large number of noise features which are seldom selected. GAMI-Net has a similar pruning step which removes less importance main-effects/interactions, so it is also less impacted by the random features. With L1 regularization, TreeNet did much better but still not as good as XGBoost.
> >
> > Regarding high order interactions, TreeNet is particularly good at capturing **pure** interactions (without main-effects) compared to XGBoost, as we see from the simulation case. This is because all parameters are optimized simultaneously, instead of optimizing one split at a time. This makes it a powerful tool for interaction filtering.
> >
> > **Question 5**: The version we use is from Kaggle website (https://www.kaggle.com/datasets/camnugent/california-housing-prices), which includes ocean proximity variable. The one in sklearn does not have ocean proximity. Both can be used, we just happen to use the one from Kaggle.
> >
> > **Question 6**: Thanks! The references for data sets have been added in the updated paper.

---

### Official Review · Reviewer_XWEf · 2025-10-31

**Soundness:** 3
**Presentation:** 3
**Contribution:** 1
**Rating:** 4
**Confidence:** 3

**Summary:**

This paper introduces Cross Spline Net (XSN), a framework that combines models like trees, MARS, and SVM into one neural network structure. By using spline transformations and cross-network layers, XSN aims to achieve flexibility, scalability, and interpretability. It fits well with current research in explainable AI and hybrid modelling.

**Strengths:**

XSN provides a unified framework that can reproduce or approximate various non-neural network models.


The proposed framework is designed to be more interpretable than traditional black-box models like XGBoost and FCNN.

**Weaknesses:**

However, XSN also faces limits such as high computational cost, sensitivity to hyper-parameters, and weaker performance on jumpy data. These issues may make it less practical for large or time-sensitive tasks.

* Although XSN improves interpretability, it can still become complex when many polynomial terms are used. The required pruning and purification steps are costly and may not always work well. Interpreting high-order interactions is also difficult, even with the purification algorithm.

* XSN may still overfit on high-dimensional data if regularisation is not carefully applied.

* While XSN uses ADAM for optimisation, the large feature space created by spline transformations can slow down training on very large datasets.

* The model assumes that tabular data has mostly low-order interactions. This may not be true for complex real-world data, which limits its general use.

* XSN introduces many hyperparameters (e.g., number of cross layers, spline bases, and projection size). These require careful tuning, which can be slow and expensive.

* The experiments only compare XSN with XGBoost and FCNN. Including other interpretable models, such as GAMI-Net, would make the evaluation more complete.

* XSN handles jumpy data less effectively than XGBoost, which works better for such patterns. This could restrict its application in scenarios where the underlying data patterns are highly discontinuous or involve sharp changes

* While the results on public datasets are promising, the paper does not clearly show how XSN performs on large, real-world problems where simpler models might still work better.

**Questions:**

How does the computational cost of the interpretability tools, e.g., pruning and purification algorithms scale with the size and complexity of the dataset?


How does the interpretability of XSN compare to other interpretable models like GAMI-Net or SHAP-based methods in terms of ease of use and insights provided?


Are there specific types of data or problems where XSN is not suitable?

---

> ### Author Response · Authors · 2025-11-21
> **Author Response Part One**
>
> We thank the reviewer for the valuable feedback. We have made significant improvement in the updated paper to address the comments.
> > XSN also faces limits such as high computational cost
>
> In the original run time table, we only reported the time per epoch/tree. In this revision, we added the total run time as this is the most relevant. FCNN and XSN take much fewer iterations to converge, since there can be 100 parameter updates in one epoch. For total run time, we can see the speed of XSN is comparable with FCNN and XGBoost for 100K case, and much faster than XGBoost in the 1M case since XGBoost requires many more iterations to converge (with a reasonable learning rate of 0.05 for XGBoost). Finally, XSN can be run with GPU to be even faster.
>
> | |         | |TreeNet| | |FCNN| | |XGBoost| |
> |-------------------|-------|-------|-------|-------|-------|-------|-------|-------|-------|---|
>  |n | p | s/iter | iter | Total | s/iter | iter | Total | s/iter | iter | Total |
> | 100K | 30   | 0.38 | 104 | 40        | 0.25 | 204 | 51      | 0.018 | 1785 | 32 |
> | 100K | 100 | 0.60 | 159 | 96      | 0.26 | 192 | 51 |  0.049 | 1170 | 57 |
> | 1M | 30     | 0.66 | 137 | 90     | 0.40 | 304 | 122 |   0.26 | 2000 | 521 |
> | 1M | 100   | 1.31  | 204 | 267 |     0.55 | 400 | 220 |  0.52 | 2000 | 1035 |
>
>  > sensitivity to hyper-parameters (HP)
>
> We added an ablation study in Appendix, where we adjust the values for 3 key HPs (learning rate, number of bases per variable and projection dimension). TreeNet is robust to HPs, with performance difference within data splitting variation on two representative real-world datasets (MSLR and HELOC). For example, for HELOC data, default TreeNet2 (bold face) has a testing AUC of 0.804, which is between the worse (0.802) and best (0.809), and the difference is smaller than the standard deviation caused by random data splitting (0.010, see Table 2 in the paper)
>
> | LR      | HELOC (AUC) | MSLR (MSE) | No. Basis | HELOC (AUC) | MSLR (MSE) | Proj. Dim | HELOC (AUC) | MSLR (MSE) |
> |---------|-------------|------------|-----------|-------------|------------|-----------|-------------|------------|
> | 0.0025  | 0.802       | 0.548      | 3         | 0.808       | 0.546      | 10        | 0.809       | 0.549      |
> | 0.005   | 0.805       | 0.546      | 4         | 0.807       | 0.545      | **20**    | **0.804**   | **0.548**  |
> | 0.01    | 0.809       | 0.545      | **5**     | **0.804**   | **0.546**  | 30        | 0.808       | 0.545      |
> | **0.02**| **0.804**   | **0.546**  | 6         | 0.804       | 0.545      | 40        | 0.809       | 0.576      |
> | 0.04    | 0.808       | 0.571      | 7         | 0.807       | 0.569      | 50        | 0.806       | 0.574      |
>
> > Weaker performance on jumpy data
>
> Agree, but XGBoost should perform best for jumpy data. Also, as sample size increase, XSN gets closer to XGBoost for jumpy case (MSE for 3way jump case reduces from 1.42 to 1.18, for 50K sample). Finally when model robustness is important, jumpiness is undesirable. Some small loss in performance is acceptable in such case.
>
>  > Although XSN improves interpretability, it can still become complex when many polynomial terms are used.
>
> > How pruning and purification algorithms scale with the size
>
> This is exactly one of our three key contributions (see “Related Work” section in the new paper). One significant gap we are filling is model interpretation after a (especially higher-order) FANOVA model is fitted. Without it, the interpretability benefit won’t be achieved. Little research is done in this area so far and we provided a systematic way to do this.
> Regarding efficiency, our method is efficient. For the 3way_cont example with 10K size, there are 585K polynomial terms. After filtering, only 20K terms exceed the conservative 0.001 L1-norm threshold, and total filtering time is just 7.8 seconds. The refitting of top functional components takes only 0.6 seconds. It leaves 14 functional components after pruning. Finally, the purification is fast given each purification model is fitted with only 2 or 3 variables. Each component took about 4 seconds and purification for components with same order is done in parallel.
> When sample size increases to 100K, filtering time only increases to 9.7 seconds, and purification time per component increases to 7 seconds. Beyond 100K, the results are very stable so a subsample suffices.
> When number of variables increases, the important effects may not grow much (otherwise, interpretation is impossible). So it does not affect purification as much. We are also testing L1 regularization to induce sparsity in coefficients.
>
> > Interpreting high-order interactions is also difficult
>
> Agree this is not easy, but this is why research needs to be done in this area. For high-order interaction, one approach often used in the industry, is to segment based on interacting variables. This way lower interaction model can be fitted within each segment.

---

> ### Author Response · Authors · 2025-11-21
> **Author Response Part Two**
>
> > XSN may still overfit on high-dimensional data
>
> XSN comes with regularization. We tested on new high-dimensional data, MSLR (1 Million observations, 136 variables) and Madelon (500 variables, 2600 observations). For MSLR, XSN does not overfit since number of observations is large. For Madelon, it has too few observations (especially after data splitting) relative to number of variables. XSN with L1 regularizer on projection layer works well. Adding regularization improves test AUC from 0.63 to 0.83 for Madelon data (see the performance table in our response to later comments).
>
> > spline transformations can slow down training
>
> The connection between input and spline layer is very sparse, since each input is only connected to its splines. For example, with 100 variables and 500 splines after transformation, the number of weight parameters between input layer and spline layer is only 500, whereas FCNN induces many more parameters. In addition, the projection layer effectively reduces dimension to only 20 for TreeNet2. This makes XSN fast as we see earlier from total run time.
>
> > The model assumes that tabular data has mostly low-order interactions
>
> This is observed in other work, like Lou et al. (2013) and Dubey et al., (2022).
>
> > XSN require careful tuning, which can be slow and expensive.
>
> We have shown in the paper that we offer an “off-the-shelf” version (TreeNet2), with default hyper parameters, that performs well in general.
>
> > Including other interpretable models
>
> > XSN on large, real-world problems
>
> We added GAMI-Net and second order TPNN in the updated Section 5, as well as two additional high-dimensional data (MSLR and Madelon). We can see TreeNet and TreeNet2 again show robust model performance with small overfitting gap. The performance is comparable with XGBoost on all datasets except for California housing and Madelon, where XGBoost is significantly better than **all other models**. For California housing, we suspect this is similar to the jumpy case in our simulation study, where the underlying data pattern is jumpy and more suitable for XGBoost model. For Madelon, this is a 'wide' data as it has many features (500) but only 2600 total observations (and training sample is even smaller after data splitting) and can overfit easily. TreeNet didn't do well initially (with test AUC around 0.63) since after spline transformation, the number of bases can exceed sample size and overfits. Therefore, we added an L0 regularizer to the projection layer, which penalizes the (thousands of) spline bases. This effectively mitigates the overfitting issue, and the model performance is much better. Note for all other datasets, regularization is not needed even for MSLR which has 136 features but sufficient sample size. Finally, comparing TreeNet with the other two interpretable models (GAMI-Net and TPNN2), TreeNet has better performance for most cases, except for Madelon where GAMI-Net is slightly better.
>
>
> | Data       | Metric     | TreeNet       | FCNN          | XGBoost       | TreeNet2      | XGBoost3      | GAMINet        | TPNN2          |
> |------------|-----------|---------------|---------------|---------------|---------------|---------------|---------------|---------------|
> | BikeShare  | Test MSE  | 0.114 (0.005) | 0.119 (0.003) | 0.104 (0.003) | 0.115 (0.004) | 0.110 (0.004) | 0.157 (0.041) | 0.190 (0.025) |
> | CalHousing | Test MSE  | 0.275 (0.012) | 0.272 (0.014) | 0.213 (0.007) | 0.276 (0.011) | 0.226 (0.007) | 0.323 (0.024) | 0.281 (0.011) |
> | MSLR       | Test MSE  | 0.543 (0.004) | 0.551 (0.001) | 0.530 (0.004) | 0.548 (0.006) | 0.548 (0.001) | 0.572 (0.005) | 0.561 (0.003) |
> | HELOC      | Test AUC  | 0.797 (0.010) | 0.790 (0.008) | 0.789 (0.009) | 0.796 (0.010) | 0.796 (0.008) | 0.799 (0.008) | 0.793 (0.009) |
> | Spambase   | Test AUC  | 0.981 (0.003) | 0.978 (0.003) | 0.986 (0.003) | 0.980 (0.003) | 0.985 (0.002) | 0.977 (0.003) | 0.974 (0.012) |
> | Magic      | Test AUC  | 0.935 (0.005) | 0.928 (0.005) | 0.933 (0.005) | 0.935 (0.004) | 0.933 (0.005) | 0.922 (0.007) | 0.907 (0.010) |
> | Madelon    | Test AUC  | 0.828 (0.022) | 0.628 (0.032) | 0.881 (0.047) | 0.819 (0.088) | 0.808 (0.022) | 0.840 (0.030) | 0.615 (0.01)  |
>
>
>  > Compare interpretation with GAMI-Net or SHAP-based methods
>
> The interpretation is similar as GAMI-Net (both based on FANOVA decomposition), but XSN is more general (higher order) and tackles the more challenging problem of interpretation without interaction filtering as a pre-processing step (interaction filtering is difficult in >=3 order). SHAP is a local explanation method, different from the global FANOVA explanation.
>
> > where XSN is not suitable
>
> It is not suitable for image or language data. As we mentioned, it is designed for tabular data.

---

### Author Response · Authors · 2025-11-21
**Overall Comment**

We sincerely thank all the reviewers for their time and valuable feedback. We have addressed all comments and made significant improvements in the paper, adding "Related Work" section to clarify the key novelties, additional interpretable models (ANOVA-TPNN [1], GAMI-Net [2]) and higher-dimensional datasets (Madelon, MSLR) in real data experiments, hyper-parameter ablation study, updated run-time comparison, etc. Here, we mainly want to **clarify the contributions of our work**, which we didn’t clearly explain initially.

Our framework solves the higher-order functional ANOVA (FANOVA) modeling and interpretation problem with a **simple, elegant and effective neural network design and a comprehensive set of model interpretation algorithms**. The core of higher-order FANOVA modeling/interpretation is the data and parameter explosion issue, given the number of possible interactions grow exponentially with the order. Our work **addresses both core issues**.

1. For the modeling issue, we propose cross spline net (XSN), a simple, elegant and effective network design which combines spline layer, projection layer, and cross layer to model a higher-order polynomial function in a scalable (where number of parameters grows only linearly) and efficient way. The spline layer provides flexibility and facilitates feature engineering, the projection layer reduces dimension to make it computationally efficient, and the cross layer engineers interactions. The three components work seamlessly together. Compared to ANOVA-TPNN and GAMI-Net (limited at second order), our method does not require an interaction filtering step to simplify the problem, plus higher-order interaction filtering itself is very challenging. We also don’t rely on low-rank matrix approximation ([3], [4]) as other polynomial models do. XSN is a novel approach that uses simple yet effective network design to solve the problem. Not only does our method perform well (better than FCNN, GAMI-Net and TPNN in most cases, and worse than XGBoost in only two cases while similar in other cases, see Table 2), it is also fast given its simple and effective design (much faster than GAMI-Net and TPNN, see Table 3). Simple and effective design makes our approach efficient, intuitive, easy to implement and adopt for large scale applications. It is also extensible where different spline transformations can be plugged in, and regularizers can be added for different components.

2. The interpretation issue is challenging due to the same parameter explosion issue. For interpretation, the fitted polynomial model must be fully expanded. This creates a potentially huge coefficient table and there is no work done to directly address this important issue (CAT [3] avoids this issue by grouping the features into a few ‘concepts’). We are filling that gap. Our pruning algorithm cleverly uses Holder’s inequality to filter each polynomial term in just O(1) complexity, which removes most of the unimportant terms efficiently. For example, using MSLR data as an example (see response part four to reviewer o2Ft), the number of coefficients after polynomial expansion is a whopping 48 million, but the filtering took less than 6 minutes to reduce it to 8234 terms (with a small L1-norm threshold 0.001). Then the pruning by functional components becomes much easier, as well as purification for top components. This shows the scale of the parameter explosion issue and the effectiveness of our proposed method.

3. Finally, we want to stress **the potential broader impact of our work**. Our work provides a new way of problem solving. As mentioned in the paper, our “unification” effort aims at solving general problems by designing appropriate neural networks that represent the problem. Researchers only need to design appropriate architecture, where the low-level optimization is taken care of by existing neural network optimization routines (which are powerful and efficient). We believe this is an elegant and efficient way of problem solving, better than ad-hoc solutions we see in the industry. In fact, as alluded to in the paper and our comments, cross spline net (with some simplifications) provides an elegant and efficient solution to the high-order interaction filtering problem (through gradient descent instead of exhaustive search which is not scalable). This solution is much needed for generalizing EBM [5], GAMI-Net to higher orders and is also required by ANOVA-TPNN.

**References**

[1] Park, Seokhun, et al. Tensor Product Neural Networks for Functional ANOVA Model. ICML 2025.

[2] Zebin Yang, et al. Gami-net: An explainable neural network based on generalized additive models with structured interactions. Pattern Recognition 2021.

[3] Viet Duong, et al. Cat: Interpretable concept-based taylor additive models. KDD ’24

[4] Abhimanyu Dubey, et al. Scalable interpretability via polynomials. NeurIPS, 2022

[5] Yin Lou, et al. Accurate intelligible models with pairwise interactions. KDD ’13

---

### Meta-Review · Area_Chair_bgyQ · 2026-01-05

**Summary:**

The main points of critics are:
- computational complexity (combinatorial explosion when many polynomial terms are used, costly pruning and purification steps, hyperparameter tuning etc.)

- Problematic interpretation of  high-order interactions, even after "purification".
- Insufficient experimental validation: missing large real-world problems with large n and p,  insufficient baseline models, etc.

- Unclear novelty compared to existing models, such as fANOVA models
- lacking theoretical guarantees regarding algorithmic convergence

- Parameter sensitivity analysis missing.

**Reviewer Concerns:**

To some degree, the questions about computational efficiency could be addressed in the rebuttal, but in my opinion,  really high-dimensional settings that would additionally require highly nonlinear models are still not considered.
- Most of the more fundamental concerns, such as unclear novelty, complicated hyperparameter tuning,  missing parameter sensitivity analysis could not be fully addressed in the rebuttal.

**Reviewer Scores:**

One reviewer already mentioned that that the score was increased (from 2 to 4), but given the many open questions, I don't see good reasons for reviewers to change their scores in a really significant way.

---

### Decision · Program_Chairs · 2026-01-26

Reject